# PERCEIVE FAST, THINK SLOW: A COGNITIVE-INSPIRED FRAMEWORK FOR TIME SERIES ANALYSIS

## ABSTRACT

Time series modeling faces persistent challenges: fixed-window tokenization misaligns with natural event boundaries, uniform computation wastes capacity on simple patterns, and static architectures cannot adapt to diverse temporal dependencies. We propose **PeCo-TS**, a cognitive-inspired framework that instantiates the principle of "perceive fast, think slow" through three key innovations: (1) *event-driven dynamic-length tokenization* that aligns tokens with semantic boundaries and reduces redundancy, (2) a *Slow–Fast dual-pathway architecture* that separates rapid perception of fine-grained variations from slower abstraction of event-level structures, and (3) *Dual-Axis Adaptive (DA$^2$) attention* that dynamically balances intra-series and inter-series dependencies via learnable gating. Extensive experiments across **four** diverse tasks—forecasting, classification, anomaly detection, and imputation—demonstrate the broad applicability of PeCo-TS, yielding consistent improvements over Transformer and linear baselines, including 5.6% lower forecasting MSE, 9.3% lower imputation error, higher classification accuracy across UCR/UEA benchmarks, and a 6.7% relative F1 gain in anomaly detection. Beyond accuracy, PeCo-TS achieves favorable efficiency–performance trade-offs by leveraging event-level abstraction and complementary pathway synergy, while its learned boundaries align with real-world regime shifts, providing interpretability. These results establish PeCo-TS as a *strong supervised architecture* with excellent per-task performance for scenarios where task-specific training data is available, offering a cognitively principled and efficient alternative to both fixed-patching Transformers and computationally heavier multi-scale approaches.

## 1 INTRODUCTION

Time series data drives critical decision-making across diverse domains including climate monitoring, energy management, financial trading, healthcare diagnostics, and industrial automation. Real-world time series exhibit rich temporal complexity: abrupt regime shifts such as market crashes or equipment failures coexist with gradual trends such as seasonal variations or long-term growth, while high-frequency noise interleaves with persistent periodic patterns such as daily cycles and weekly rhythms. To effectively support the growing spectrum of tasks, including forecasting future values, classifying temporal patterns, detecting anomalies, and imputing missing data, models must capture both transient events and long-term dependencies across multiple temporal scales.

Despite this complexity, most approaches still follow a rigid three-stage pipeline. First, they split a series into fixed-size patches and treat each patch as a token. Second, a uniform architecture (e.g., self-attentive Transformer or MLP) assigns the same amount of compute to every token. Third, task heads project hidden states to outputs (e.g., forecasting, classification, anomaly detection). While convenient, this recipe clashes with heterogeneous real-world signals and leads to three limitations: (i) *boundary misalignment*—fixed windows cut through meaningful events (e.g., crashes, daily cycles, anomaly onsets), yielding incoherent representations (Nie et al., 2023; Wu et al., 2023); (ii) *computational redundancy*—expensive attention is spent on simple trends while complex patterns remain under-modeled (Zeng et al., 2023; Chen et al., 2023); and (iii) *limited adaptivity*—static channel handling cannot balance intra-series temporal dependencies against inter-series cross-channel correlations (Zhou et al., 2023; Han et al., 2023).

Cognitive neuroscience provides a useful blueprint. Human perception operates through dual pathways: *fast perceptual streams* that capture high-frequency details for immediate responsiveness, and

*slower integrative streams* that abstract low-frequency regularities into coherent events and higher-level concepts (Zacks and Swallow, 2007; Kahneman, 2011; Desimone and Duncan, 1995; Kiebel et al., 2008). Crucially, the brain performs *adaptive event segmentation*, partitioning continuous inputs into variable-length events such as daily cycles, regime changes, or anomaly onsets, rather than rigid temporal windows (Zacks and Swallow, 2007). Higher-order processing further leverages *selective attention*, shifting focus between temporal patterns within streams and cross-modal correlations across channels (Grondin, 2010). Together, these mechanisms concentrate computation on meaningful units while maintaining efficiency through event-level abstraction.

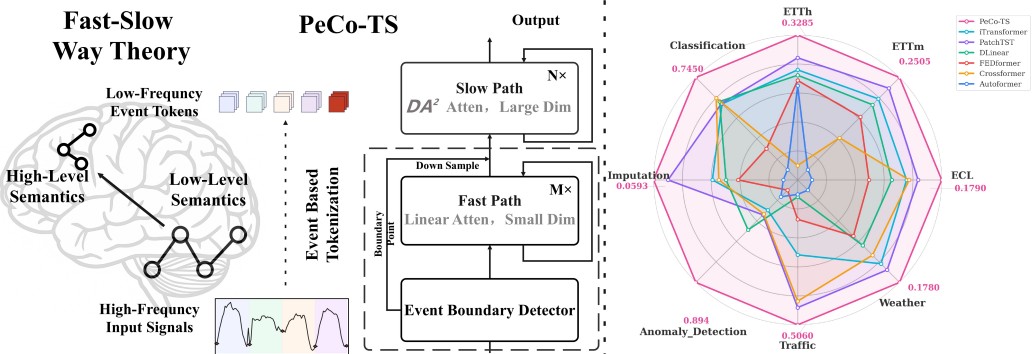

Figure 1: **Overview and highlights.** *Left:* Cognitive motivation and architecture of **PeCo-TS**, which integrates event-driven segmentation, a Fast Path for high-frequency perception, a Slow Path for event-level abstraction, and $DA^2$ attention for adaptive dependency modeling. *Right:* Aggregated results across four core time-series tasks show that PeCo-TS achieves consistent accuracy gains and superior performance compared to strong baselines.

Inspired by cognitive neuroscience, we propose **PeCo-TS** (Perception–Concept Transformer for Time Series), a dual-pathway framework that couples rapid perception with slower conceptual abstraction (Figure 1, left). The *Fast Path* employs point-wise embedding and linear attention to capture high-frequency details and transient events. An *event-driven tokenization* module, guided by frequency-domain boundary detection, adaptively segments sequences into variable-length tokens aligned with intrinsic dynamics. These tokens are then processed by the *Slow Path* using Dual-Axis Adaptive ($DA^2$) attention, which balances temporal dependencies within each series and cross-channel correlations via a learnable gating mechanism. In this way, PeCo-TS replaces rigid fixed-window patching with adaptive event segmentation, reduces computational complexity through event-level abstraction, and allocates attention more effectively while preserving both local fidelity and global coherence.

Comprehensive experiments confirm the advantages of this cognitively inspired design (Figure 1, right). Across **four** core time-series tasks (forecasting, classification, anomaly detection, and imputation), PeCo-TS consistently outperforms state-of-the-art Transformer and linear baselines while offering superior accuracy–efficiency trade-offs. Furthermore, the learned event boundaries align well with real regime shifts and anomalies, providing intuitive insights into temporal dynamics and validating the semantic relevance of our adaptive segmentation. Positioning PeCo-TS as a *strong supervised architecture* optimized for per-task performance (rather than a foundation model requiring large-scale pretraining), our key contributions are threefold: (1) a novel **event-driven dynamic-length tokenization** framework that fundamentally replaces fixed-window patching with boundary-aware segmentation, achieving $4.8\times$ token compression while preserving semantic coherence; (2) a **Slow–Fast dual-pathway architecture** that separates rapid perception from conceptual abstraction, mirroring the brain's perceive-fast, think-slow strategy with $1.85\times$ faster inference and 28% lower memory; and (3) a **Dual-Axis Adaptive ($DA^2$) attention** mechanism that dynamically balances intra-series and inter-series dependencies through learnable gating, demonstrating consistent state-of-the-art performance across four diverse time-series tasks.

## 2 RELATED WORK

**Adaptive Tokenization and Multi-Resolution Modeling.** While fixed-size patching (PatchTST (Nie et al., 2023), TimesNet (Wu et al., 2023)) remains dominant, recent work explores adaptive alternatives. MultiResFormer (Peršak et al., 2024) uses multiple fixed resolutions,

DeformableTST (Luo and Wang, 2024) adapts attention spans, and token merging (Götz et al., 2024) post-hoc merges existing tokens. Lightweight alternatives (Linear (Zeng et al., 2023), TSMixer (Chen et al., 2023)) reveal redundancy in uniform Transformers. Crucially, these differ from our approach: MultiResFormer requires predetermined scales; token merging operates post-hoc rather than learning boundaries from signal structure; deformable attention adjusts spans but not tokenization itself. Our learnable event-driven segmentation replaces fixed windows with frequency-guided boundaries that adapt end-to-end, providing semantically coherent tokenization with reduced cost. Section 4 provides controlled ablations against fixed patching and token merging baselines.

**Foundation Models and Multi-Task Learning.** Large-scale pre-trained models (TimesFM, Chronos, MOIRAI (Das et al., 2024; Shchur et al., 2024; Bhatnagar et al., 2024)) leverage diverse data for zero-shot generalization, few-shot learning, and cross-task transfer. Notably, UniTS (Gao et al., 2024) unifies predictive and generative tasks through task tokenization, achieving strong performance across 38 datasets—demonstrating the value of unified multi-task architectures. These foundation models excel at data efficiency and representational reuse but require extensive pretraining on large-scale corpora. iTransformer (Zhou et al., 2023) models variables as tokens, while MCformer (Han et al., 2023) dynamically groups channels. PeCo-TS addresses a *complementary* direction: rather than competing with foundation models on transfer learning, we focus on *per-task supervised learning* where task-specific data is available, achieving superior accuracy through cognitively inspired architecture design without requiring large-scale pretraining. Our $DA^2$ attention adaptively balances intra- and inter-series correlations, outperforming both channel-independent and channel-dependent baselines.

**State-Space Models and Hybrid Architectures.** State-space models (SSMs) like Mamba have gained prominence for linear-time complexity in long sequences. TimeMachine (Ahamed and Cheng, 2024) applies Mamba to time series with superior scalability and memory efficiency, while SST (Xu et al., 2025) (CIKM 2025) introduces a hybrid Mamba-Transformer architecture with expert modules that separate long-range patterns (Mamba) from short-term dynamics (Transformer). Diffusion models (DyDiff (Guo et al., 2025)) model temporal transitions for spatiotemporal prediction. Cognitive dual-pathway processing (Zacks and Swallow, 2007; Feichtenhofer et al., 2019) motivates "perceive fast, think slow" designs. PeCo-TS differs fundamentally: unlike SSMs operating uniformly across timesteps or fixed-rate dual pathways, we integrate *learnable event-driven segmentation* with adaptive Fast-Slow processing where boundaries are end-to-end trainable and adapt to signal-specific rhythms, explicitly separating perception from abstraction with event-level efficiency.

## 3 METHODOLOGY

### 3.1 OVERVIEW OF PECO-TS

The human brain processes continuous sensory streams through a dual-pathway system: a *fast pathway* that responds rapidly to fine-scale stimuli, and a *slow pathway* that integrates information over longer horizons to form abstract concepts. This division of labor allows cognition to capture both transient details and stable regularities. In contrast, existing Transformers for time series typically rely on a single processing pipeline with fixed patching and uniform attention, which fails to reflect the heterogeneous timescales and adaptive correlations inherent in real signals.

Inspired by this neuro-cognitive principle, we propose the **Perception–Concept Transformer for Time Series (PeCo-TS)**, a dual-pathway architecture designed to model event-driven signals with both efficiency and accuracy (see Figure 2). The framework integrates four coordinated stages: (i) *Event Boundary Detector* that identifies semantic boundaries for adaptive tokenization; (ii) *Fast Path* that captures fine-grained details through point-wise processing, followed by a segmentation-and-downsampling step that converts high-resolution features into event-level tokens; (iii) *Slow Path* with $DA^2$ attention that processes these event-based tokens for abstract modeling; and (iv) *Temporal Reprojection* that fuses abstract and fine-grained representations for multi-task outputs.

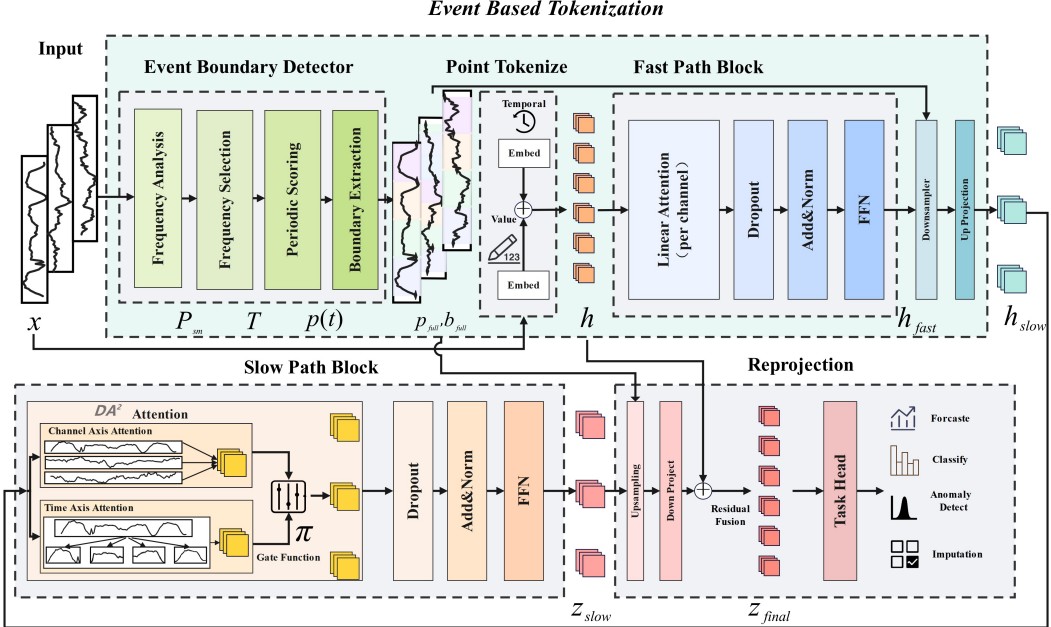

Figure 2: **Overall framework of PeCo-TS.** The framework integrates four coordinated stages: Event Boundary Detector identifies semantic boundaries for adaptive tokenization; Fast Path captures fine-grained details through point-wise processing and linear attention; Slow Path processes event-level tokens with DA$^2$ attention for adaptive intra- and inter-series dependencies; and Temporal Reprojection fuses abstract and fine-grained representations for multi-task outputs.

## 3.2 EVENT BOUNDARY DETECTOR

Modeling long sequences with uniform patches is not only computationally expensive but also misaligned with the event-driven nature of real signals. In practice, important transitions often occur at irregular intervals, making fixed patching prone to cutting through meaningful events. To address this, we design an **event-driven tokenization module** that detects semantic boundaries directly from the raw multivariate input $x \in \mathbb{R}^{B \times L \times C}$, ensuring that subsequent processing aligns with natural temporal structure.

For each channel, the dominant rhythm is estimated by computing the power spectrum and applying a learnable frequency smoother $g_\theta$:

$$P = |X|^2, \quad X = \text{FFT}(x), \quad P_{\text{sm}} = g_\theta(P). \tag{1}$$

A softmax distribution with temperature $\tau$ softly selects frequency bins to obtain an effective frequency $k_{\text{eff}}$ and period $T$:

$$\alpha = \text{softmax}\left(\frac{P_{\text{sm}}}{\tau}\right), \quad k_{\text{eff}} = \sum_f \alpha_f \cdot f, \quad T = \frac{L}{k_{\text{eff}}}. \tag{2}$$

A differentiable cosine comb then highlights candidate boundaries:

$$p(t) = \left(\frac{1+\cos(2\pi t/T)}{2}\right)^\gamma, \quad \gamma > 1, \tag{3}$$

where the learnable sharpness $\gamma$ adjusts boundary precision. Non-maximum suppression and thresholding convert these scores into binary boundaries $b_{\text{full}} \in \{0,1\}^{B \times L \times C}$, while soft probabilities $p_{\text{full}} \in [0,1]^{B \times L \times C}$ are retained.

This mechanism aligns tokenization with the inherent rhythm of each channel, yielding three advantages: (*i*) the number of tokens adapts to signal-specific periodicity; (*ii*) the boundaries are differentiable and trainable, enabling end-to-end optimization; and (*iii*) by operating on event tokens rather than all time steps, the complexity of later attention layers is reduced from $O(L^2)$ to $O(M^2)$ with $M \ll L$.

### 3.3 FAST PATH: PERCEPTION OF FINE DETAILS

While boundaries guide event-level abstraction, retaining fine-grained local details remains essential for accurate modeling. Analogous to early cortical areas in human perception, the **Fast path** processes the input at its original resolution to preserve high-frequency variations and transient patterns.

Formally, each scalar observation $x_{t,c}$ is embedded into a $d_f$-dimensional vector via a point-wise tokenizer:

$$h \in \mathbb{R}^{B \times L \times C \times d_f}. \tag{4}$$

After reshaping and adding positional encodings, we obtain $h \in \mathbb{R}^{BC \times L \times d_f}$, which is then processed with *linear attention* to efficiently capture short-range dependencies:

$$\text{Attn}(Q, K, V) \approx \frac{\phi(Q)\left(\phi(K)^\top V\right)}{\phi(Q)\,\phi(K)^\top}, \tag{5}$$

reducing time complexity from $O(L^2)$ to $O(Ld)$.

The resulting representation $h_{\text{fast}}$ preserves temporal precision and is projected into a higher-dimensional space $h_{\text{fast}} \in \mathbb{R}^{BC \times L \times d_h}$. Guided by the boundaries $b_{\text{full}}$ from Section 3.2, a boundary-aware downsampler aggregates $h_{\text{fast}}$ into variable-length event tokens:

$$h_{\text{slow}} = \text{Downsample}(h_{\text{fast}}, b_{\text{full}}) \in \mathbb{R}^{B \times C \times M \times d_h}, \quad M \ll L. \tag{6}$$

Since different channels may yield different token counts $M_c$, we pad sequences to $M_{\max} = \max_c M_c$ and maintain a mask $\mu \in \{0,1\}^{B \times C \times M_{\max}}$ to ensure consistent computation. This design enables the model to preserve fine details while seamlessly transitioning to event-level abstraction.

### 3.4 SLOW PATH: CONCEPTUAL ABSTRACTION

High-level perception in the brain does not stop at detecting local events; it further integrates them into coherent concepts by linking information across time and across modalities. Following this principle, the **Slow path** in PeCo-TS takes event-level tokens as input and abstracts them into higher-order representations using a dual-axis adaptive attention mechanism.

Formally, given event tokens $h_{\text{slow}} \in \mathbb{R}^{B \times C \times M \times d_h}$ and mask $\mu$ (with $M$ denoting $M_{\max}$), DA$^2$ attention decomposes modeling into two complementary axes. Along the *token axis*, attention captures temporal dependencies across events within each channel. Along the *channel axis*, attention captures correlations across channels at the same event step. Padded positions are excluded using $\mu$ (see Appendix A.5):

$$\tilde{z}_c(b, c, \cdot) = \text{Attn}_{\text{token}}\big(h_{\text{slow}}(b, c, \cdot, \cdot)\big) \in \mathbb{R}^{M \times d_h}, \tag{7}$$

$$\tilde{z}_m(b, \cdot, m) = \text{Attn}_{\text{channel}}\big(h_{\text{slow}}(b, \cdot, m, \cdot)\big) \in \mathbb{R}^{C \times d_h}. \tag{8}$$

Both outputs are reshaped to a common layout and blended by a learnable gate $\pi \in (0, 1)$:

$$Y = \pi \odot \tilde{z}_m + (1 - \pi) \odot \tilde{z}_c \in \mathbb{R}^{B \times C \times M \times d_h}. \tag{9}$$

Unless otherwise specified, $\pi$ is a per-layer scalar broadcast as $B \times C \times M \times 1$, balancing inter-series and intra-series modeling. A finer variant allows per-position gating $\pi \in (0, 1)^{B \times C \times M \times 1}$, but we use the scalar form by default for stability.

Stacking multiple DA$^2$ layers with residual and feedforward modules produces the abstract representation $z_{\text{slow}} \in \mathbb{R}^{B \cdot C \times M \times d_h}$, which jointly encodes long-horizon temporal dependencies and context-dependent cross-channel relations. This abstraction is particularly important for multivariate event-driven time series, where both within-series evolution and cross-series interactions carry critical semantics (see Appendix A.3).

### 3.5 TEMPORAL REPROJECTION AND MULTI-TASK HEADS

Event tokens are efficient for abstraction but not directly aligned with the fine temporal resolution required by downstream tasks. To bridge this gap, we design a **temporal reprojection layer** that up-samples event-level features back to the original scale, restoring temporal alignment while injecting high-level semantics.

Given $z_{\text{slow}} \in \mathbb{R}^{B \cdot C \times M \times d_h}$ and boundary indicators $(p_{\text{full}}, b_{\text{full}})$, the reprojection constructs convex weights $\{w_{t,i}\}_{i=1}^{M}$ for each time step $t$:

$$z_{\text{full}}(t) = \sum_{i=1}^{M} w_{t,i} \, z_{\text{slow}}(i), \quad \sum_{i=1}^{M} w_{t,i} = 1. \tag{10}$$

Segments $\mathcal{S}_i = [s_i, e_i]$ are defined by consecutive boundaries in $b_{\text{full}}$. Within each segment, unnormalized weights are assigned as $\tilde{w}_{t,i} = \kappa(\text{dist}(t; s_i, e_i)) \, \bar{p}(t)$, where $\bar{p}(t)$ is the channel-aggregated confidence from $p_{\text{full}}$ and $\kappa(d) = \exp(-d^2/2\sigma^2)$ is a Gaussian kernel. Normalization yields

$$w_{t,i} = \frac{\tilde{w}_{t,i}}{\sum_{j=1}^{M} \tilde{w}_{t,j}}, \quad w_{t,i} = 0 \text{ if } t \notin \mathcal{S}_i. \tag{11}$$

Finally, the reprojected features are aligned with fast-path representations via a learnable output projection and residual fusion:

$$z_{\text{final}}(t) = W_{\text{out}} \, z_{\text{full}}(t) + h_{\text{fast}}(t), \quad W_{\text{out}} \in \mathbb{R}^{d_f \times d_h}. \tag{12}$$

The unified representation $z_{\text{final}} \in \mathbb{R}^{B \cdot C \times L \times d_f}$ forms a shared basis for diverse tasks—classification, imputation, anomaly detection, forecasting, and pretraining. This feedback from abstraction to detail resembles *predictive coding*, ensuring that conceptual modeling remains consistent with fine-grained temporal alignment (see Appendix A.6).

## 4 EXPERIMENTS

We evaluate **PeCo-TS** on four fundamental time-series tasks—forecasting, classification, anomaly detection, and imputation—using widely adopted benchmarks: forecasting on ETTh1/h2, ETTm1/m2, Electricity, Exchange, Traffic, and Weather (Zhou et al., 2021; Trindade, 2015; Lai et al., 2017; Lai and contributors, 2017; Li et al., 2018; for Biogeochemistry , data origin; Wang et al., 2024); classification on seven UCR/UEA datasets (Chen et al., 2015; Bagnall et al., 2018); anomaly detection on MSL, PSM, SMAP, SMD, and SWAT (Hundman et al., 2018; Abdulaal et al., 2021; Su et al., 2019; Goh et al., 2016); and imputation on ETTh/ETTm/Electricity/Weather. This comprehensive evaluation setting ensures coverage of both short- and long-horizon prediction, univariate and multivariate inputs, and diverse application domains.

### 4.1 BROAD APPLICABILITY VALIDATED BY MULTI-TASK RESULTS

We compare against diverse baselines: recent Transformers (AMD (Hu et al., 2025), Path-Former (Chen et al., 2024), CARD (Xue et al., 2024)), unified multi-task models (UniTS (Gao et al., 2024)), established baselines (iTransformer (Zhou et al., 2023), PatchTST (Nie et al., 2023), TimesNet (Wu et al., 2023)), and efficient alternatives (TSMixer (Chen et al., 2023), DLinear (Zeng et al., 2023), Mamba (Gu and Dao, 2023)).

Across all four tasks, PeCo-TS consistently outperforms strong baselines. In forecasting, it achieves 5.6% lower MSE on average (Table 1; Appendix, Table 3). PeCo-TS outperforms AMD on 6/8 datasets and matches or exceeds PathFormer, CARD, and UniTS. In classification, it surpasses leading alternatives (Table 4). For anomaly detection, F1 improves from 0.837 to 0.893 (6.7% gain, Table 5); imputation error drops 9.3% (Table 6). These consistent improvements validate PeCo-TS as a versatile backbone for time-series applications.

### 4.2 ADVANTAGES OVER FIXED PATCHING

A key limitation of conventional Transformers for time series lies in their rigid fixed-window tokenization, which fragments signals and often cuts through natural temporal boundaries. In contrast, our learnable, event-driven segmentation produces variable-length tokens that adapt to intrinsic rhythms, such as daily cycles or volatility bursts, thereby aligning representation with the underlying event structure.

To validate its effectiveness, we compare our segmentation against fixed-patch baselines across two complementary dimensions: prediction horizon and input length. As shown in Figure 3a, event-driven segmentation consistently achieves lower MSE across horizons, with relative gains ranging

Table 1: Multivariate forecasting results with prediction lengths $S \in \{96, 192, 336, 720\}$ for all datasets and fixed lookback length $T = 96$. Results are averaged across prediction lengths. The best results are highlighted in **red** and the second best are shown in **blue**.

| Dataset | PeCo-TS | | AMD | | PathFormer | | CARD | | UniTS | | iTransformer | | PatchTST | | TSMixer | | TimesNet | | Mamba | | DLinear | |
|---|---|---|---|---|---|---|---|---|---|---|---|---|---|---|---|---|---|---|---|---|---|---|
| | MSE | MAE | MSE | MAE | MSE | MAE | MSE | MAE | MSE | MAE | MSE | MAE | MSE | MAE | MSE | MAE | MSE | MAE | MSE | MAE | MSE | MAE |
| ETTh1 | 0.415 | 0.427 | 0.435 | 0.428 | 0.445 | 0.426 | 0.442 | 0.428 | 0.454 | 0.459 | 0.465 | 0.455 | 0.448 | 0.446 | 0.626 | 0.588 | 0.460 | 0.455 | 0.550 | 0.509 | 0.460 | 0.457 |
| ETTh2 | 0.378 | 0.400 | 0.383 | 0.402 | 0.389 | 0.419 | 0.396 | 0.427 | 0.415 | 0.422 | 0.383 | 0.407 | 0.381 | 0.408 | 2.025 | 1.194 | 0.409 | 0.425 | 0.443 | 0.441 | 0.564 | 0.519 |
| ETTm1 | 0.369 | 0.388 | 0.390 | 0.400 | 0.400 | 0.403 | 0.401 | 0.413 | 0.407 | 0.413 | 0.407 | 0.411 | 0.386 | 0.400 | 0.529 | 0.513 | 0.412 | 0.418 | 0.498 | 0.468 | 0.404 | 0.408 |
| ETTm2 | 0.287 | 0.333 | 0.295 | 0.348 | 0.303 | 0.349 | 0.293 | 0.343 | 0.443 | 0.407 | 0.291 | 0.334 | 0.285 | 0.330 | 1.030 | 0.753 | 0.294 | 0.332 | 0.377 | 0.380 | 0.355 | 0.401 |
| Electricity | 0.201 | 0.282 | 0.225 | 0.310 | 0.218 | 0.315 | 0.216 | 0.300 | 0.217 | 0.317 | 0.225 | 0.308 | 0.210 | 0.297 | 0.233 | 0.340 | 0.297 | 0.376 | 0.209 | 0.311 | 0.225 | 0.319 |
| Exchange | 0.387 | 0.418 | 0.408 | 0.428 | 0.557 | 0.477 | 0.367 | 0.414 | 0.567 | 0.491 | 0.364 | 0.407 | 0.369 | 0.407 | 0.539 | 0.590 | 0.406 | 0.439 | 0.693 | 0.555 | 0.339 | 0.413 |
| Traffic | 0.527 | 0.337 | 0.586 | 0.371 | 0.544 | 0.351 | 0.535 | 0.347 | 0.543 | 0.352 | 0.612 | 0.404 | 0.526 | 0.339 | 0.606 | 0.407 | 0.903 | 0.523 | 0.679 | 0.381 | 0.672 | 0.419 |
| Weather | 0.260 | 0.283 | 0.249 | 0.279 | 0.297 | 0.310 | 0.300 | 0.311 | 0.288 | 0.302 | 0.267 | 0.287 | 0.260 | 0.281 | 0.243 | 0.309 | 0.262 | 0.288 | 0.295 | 0.315 | 0.265 | 0.317 |

Table 2: Hardware efficiency metrics across sequence lengths on ETTh1 (batch size 32, RTX 3090). Event-driven segmentation achieves effective compression with M/L $< 0.07$ across all settings.

| Seq Length (L) | Latency (ms) | Memory (MB) | Tokens (M) | M/L Ratio |
|---|---|---|---|---|
| 96 | 31.30 | 139 | 5.57 | 0.058 |
| 192 | 62.59 | 215 | 12.43 | 0.065 |
| 384 | 70.80 | 406 | 19.29 | 0.050 |
| 768 | 76.80 | 640 | 33.00 | 0.043 |
| 1536 | 85.75 | 1539 | 74.14 | 0.048 |

from 4.7% on Weather to 7.3% on ETTm1. Figure 3b further confirms robustness under varying input sequence lengths: our method maintains superior performance regardless of the temporal context size. Notably, the advantage of event-driven segmentation becomes more pronounced as input or prediction length increases. Short patches tend to split coherent events into fragments and introduce redundant tokens into higher layers, leading to inefficiency, while long patches often merge multiple events into a single token, causing semantic overlap and learning difficulty. By contrast, event-driven segmentation preserves semantic integrity within tokens while maintaining computational efficiency, thereby scaling gracefully with longer horizons and context windows.

Together, these results provide strong evidence that event-driven segmentation fundamentally improves over arbitrary fixed patches. Qualitative visualizations in the Appendix (Figures 9–15) further show that learned boundaries align with key temporal events, yielding semantically coherent and generalizable representations. This alignment underpins the consistent quantitative gains observed across datasets, establishing event-driven tokenization as a principled foundation for time-series modeling.

### 4.3 SYNERGISTIC ROLES OF COGNITIVE FAST AND SLOW PATHS

Inspired by the brain's dual-pathway system, PeCo-TS explicitly separates rapid perception from slower conceptual abstraction. To assess the necessity of this design, we perform ablation studies by removing either the Fast or Slow path. As shown in Figure 3c,d, eliminating the Fast path leads to an average 10.0% drop in MSE improvement, while removing the Slow path results in a 4.6% reduction. This asymmetric degradation underscores their complementary functions: the Fast path preserves high-frequency details crucial for precise temporal alignment (e.g., anomaly detection), whereas the Slow path processes event-level tokens to capture long-range dependencies efficiently and allocate modeling capacity to complex structures.

Beyond accuracy, this division of labor also contributes to efficiency. Table 2 quantifies the hardware efficiency across varying sequence lengths on ETTh1. The M/L ratio remains consistently low (0.04–0.07), confirming that event-driven segmentation effectively compresses sequences: the Slow path processes only 5–7% of the original timesteps, directly reducing computational complexity. Despite longer sequences requiring more memory and computation, the inference latency scales gracefully, and the efficient compression combined with lightweight FFT-based boundary detection enables PeCo-TS to attain competitive efficiency compared with strong baselines (Appendix A.14).

To further understand the computational distribution, Figure 4 presents a component-level profiling of inference time. The Fast path dominates computation (46–65%) when processing the full sequence, while the Slow DA² attention contributes 29–42% depending on token count M. Critically, the FFT-based segmentation accounts for merely 5–9% of total time, validating our design choice of operating attention on compressed event tokens rather than raw sequences. Together, these

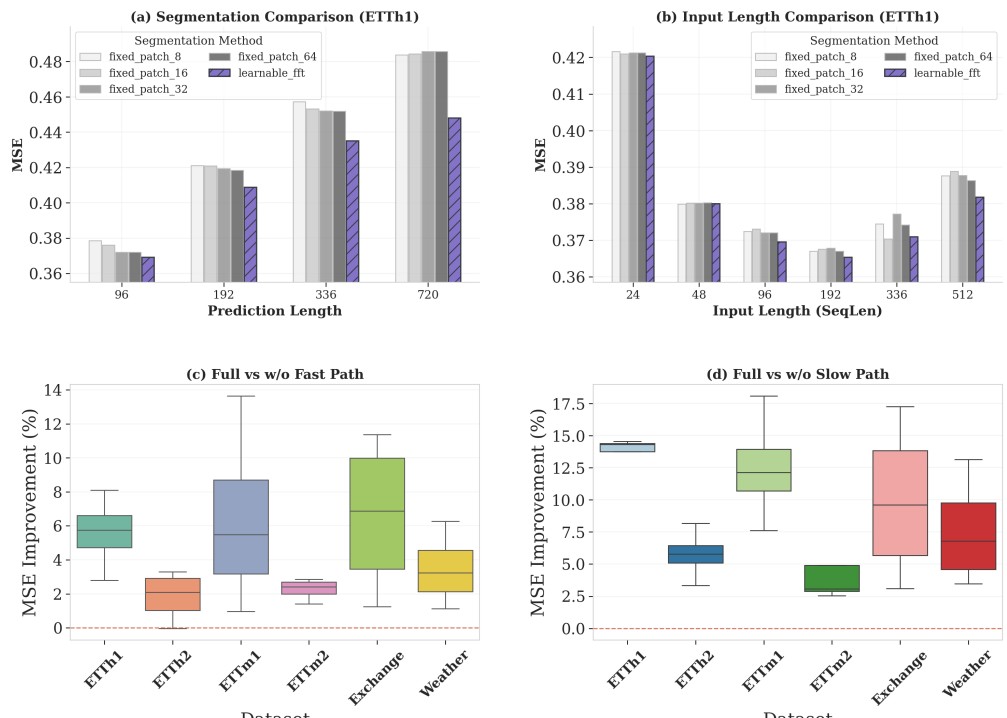

Figure 3: **Comprehensive analysis of PeCo-TS key components.** (a) **Segmentation method comparison (ETTh1):** Adaptive event-driven segmentation consistently outperforms fixed patches across prediction horizons, with learnable FFT achieving the lowest MSE. (b)**Input length sensitivity analysis (ETTh1):** Performance comparison across varying input sequence lengths and fixed 96 prediction horizon demonstrates the robustness of learnable FFT segmentation, maintaining superior performance regardless of input length variations. (c)(d) **Dual-pathway architecture validation:** Comprehensive ablation study demonstrates the complementary nature of Fast and Slow pathways. The asymmetric contributions validate our cognitive-inspired hypothesis that rapid perception and slower reasoning serve distinct but synergistic roles in time series modeling.

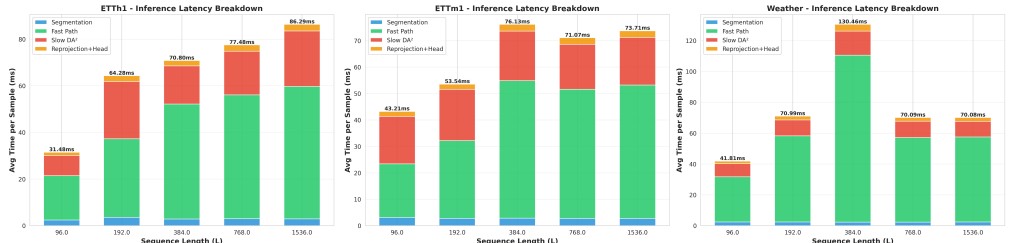

Figure 4: **Component-level inference time breakdown across sequence lengths.** Profiling on ETTh1 reveals that the Fast path dominates computation for shorter sequences, while the Slow DA² attention and segmentation maintain controlled overhead. The FFT-based boundary detection consistently accounts for only 5–9% of total time, demonstrating its efficiency.

empirical savings align with the theoretical benefits of event-driven compression and the practical synergy of the dual pathways, validating our cognitive-inspired hypothesis: rapid perception and slower abstraction are distinct yet synergistic mechanisms that jointly yield a more effective and efficient time-series model.

## 4.4 ADAPTIVE INTRA- AND INTER-SERIES DEPENDENCY MODELING

A distinctive advantage of PeCo-TS lies in its DA$^2$ attention, which adaptively balances intra-series and inter-series dependencies rather than committing to fixed channel-independent or channel-

dependent designs. As shown in Appendix Figure 23, DA$^2$ consistently outperforms both alternatives across benchmarks, with the largest margin on ETTh2 (5.2% average MSE reduction).

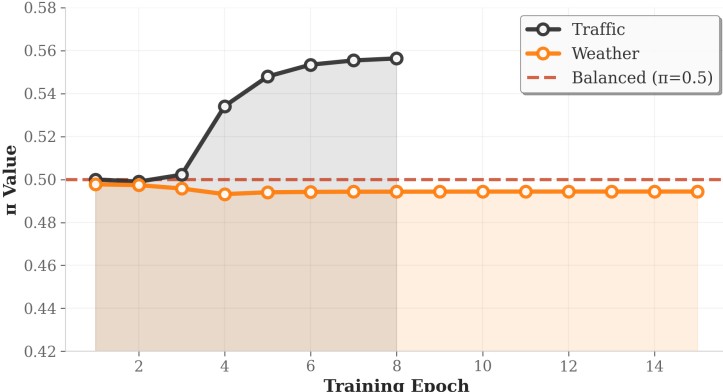

Figure 5: **Evolution of gating parameter $\pi$ across datasets.** The learnable gate parameter $\pi$ in DA$^2$ attention adapts to dataset characteristics: shifting toward intra-series modeling (lower $\pi$) for periodic data like Traffic, and toward inter-series modeling (higher $\pi$) when cross-channel correlations dominate as in Weather. This demonstrates the model's ability to dynamically balance the two attention axes based on signal properties.

Beyond accuracy, DA$^2$ attention dynamically adjusts its gate parameter $\pi$ to reflect the correlation structure of each dataset. Figure 5 illustrates that, as training progresses, the model gradually learns dataset-specific dependency patterns: on Traffic (Chen et al., 2025), where intra-series periodicity dominates, DA$^2$ increases its emphasis on intra-series attention ($\pi \approx 0.56$); on Weather (Chen et al., 2025), where cross-channel correlations are stronger, the model assigns greater weight to inter-series attention ($\pi \approx 0.49$). This adaptive learning process improves predictive accuracy while offering transparent insights into dataset-specific structures.

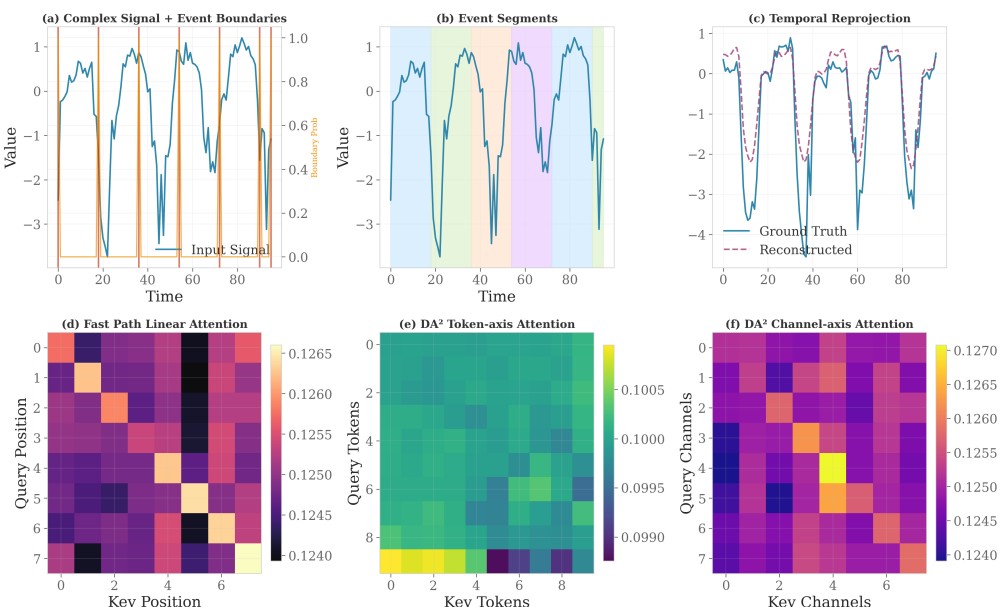

Figure 6: **Cognitive architecture visualization showing the dual-pathway processing.** (a) Input signal with learned event boundaries; (b) Event segments derived from boundary detection; (c) Temporal reprojection reconstructing fine-grained outputs from event-level abstractions; (d) Fast path linear attention exhibiting local temporal dependencies; (e) DA$^2$ token-axis attention for intra-series modeling; (f) DA$^2$ channel-axis attention for inter-series correlations.

### 4.5 Cognitive Pathway Behavior and Mechanistic Insights

The cognitive principle of "perceive fast, think slow" is instantiated in PeCo-TS through observable modeling behavior (detailed visualizations in Appendix A.1). The boundary detector first converts continuous signals into event-aligned tokens, ensuring semantic integrity at the token level. These tokens then flow into two complementary pathways: the Fast path applies linear attention with strong near-diagonal focus, retaining fine-grained local dependencies crucial for precise temporal alignment; the Slow path employs $DA^2$ attention across tokens and channels, integrating long-range structures and cross-series correlations. Temporal reprojection feeds event-level abstractions back to the native resolution, enabling consistent reconstruction of high-frequency detail.

The attention patterns provide direct evidence for this division of labor: the Fast path concentrates on short-range patterns, while the Slow path distributes capacity over broader token and channel contexts. Moreover, the gate parameter $\pi$ adapts smoothly across datasets (see Appendix Figure 5), shifting emphasis toward intra-series dependencies in periodic data (e.g., Traffic) and toward inter-series dependencies when cross-channel correlations dominate (e.g., Weather). This dynamic reallocation reflects the model's ability to specialize its reasoning strategy to the dataset at hand. Beyond dataset-level trends, layer-wise analysis reveals balanced pathway utilization: across all layers and datasets, $\text{Mean}(\pi) \approx 0.50$ with standard deviation $< 0.003$, and gating entropy reaches $H(\pi) = 0.693 \approx \ln(2)$ (maximum uncertainty), indicating the model actively leverages both branches rather than collapsing to a single pathway. Forced ablation experiments (setting $\pi = 0$ or $\pi = 1$) result in 1.1–5.8% performance degradation across datasets, validating that both Fast and Slow pathways provide non-redundant contributions essential for optimal performance.

These mechanistic observations align closely with the empirical results. Event-driven segmentation explains why PeCo-TS maintains stronger margins at longer horizons (Figure 3a,b); the complementary Fast/Slow contributions account for the asymmetric error increases in ablation (Figure 3c,d); and $DA^2$ attention clarifies why adaptive correlation modeling consistently outperforms fixed channel-independent/dependent baselines. Even under anomaly detection and missing-data scenarios, the synergy holds: slow abstractions provide contextual guidance, while fast features anchor precise timing, yielding improved localization and robustness. Together, this mechanism–phenomenon–result loop demonstrates how event alignment and dual-path reasoning shape transparent attention geometry, which directly underpins the multitask gains and favorable accuracy–efficiency trade-offs observed in PeCo-TS.

## 5 Conclusion

This work introduced **PeCo-TS**, a cognitive-inspired framework that translates the principle of "perceive fast, think slow" into a practical architecture for time series modeling. By coupling event-driven tokenization, a dual-pathway design, and $DA^2$ adaptive attention, PeCo-TS directly addresses the long-standing limitations of fixed-window segmentation, uniform computation, and static channel mixing. Extensive experiments across forecasting, classification, anomaly detection, and imputation confirm its advantages: event-driven segmentation scales gracefully with horizon and context length, the Fast and Slow paths contribute complementary precision and abstraction, and $DA^2$ attention adapts to dataset-specific dependency structures.

We note that PeCo-TS is currently designed as a *supervised architecture* for per-task performance when sufficient training data is available, rather than a foundation model supporting few-shot learning or zero-shot generalization. This positions PeCo-TS as complementary to pretrained models like UniTS and TimesFM, excelling in scenarios where task-specific data is abundant.

Looking forward, we aim to extend the cognitive dual-pathway architecture toward foundation model capabilities—exploring large-scale pretraining strategies that leverage event-driven representations for cross-domain transfer, few-shot adaptation, and unified multi-task learning. We believe this cognitive–computational synthesis provides a promising pathway toward scalable, transparent, and generalizable time-series foundation models.

ACKNOWLEDGMENTS

We thank the reviewers for their constructive feedback and suggestions. This work was supported by research grants and computational resources that enabled comprehensive experimental validation.

ETHICS STATEMENT

This research was conducted in strict compliance with ethical standards. The datasets used in our experiments are all publicly available benchmarks or synthetically generated signals without any personally identifiable or sensitive information. No human or animal subjects were involved. All experimental protocols respect the principles of fairness, transparency, and scientific integrity. The proposed methods are intended solely for academic research purposes and do not pose foreseeable risks of harm or misuse.

REPRODUCIBILITY STATEMENT

We have taken concrete steps to ensure the reproducibility of our results.

- **Code and Models:** The full implementation of our PeCo-TS architecture, including the segmentation module, DA$^2$-Attention mechanism, and experimental pipelines, is available at the following anonymous repository: https://anonymous.4open.science/r/PeCO-TS-Code-102C

- **Datasets:** All datasets used are standard public benchmarks (ETT, Electricity, Exchange, Traffic, Weather, UCR/UEA, MSL, PSM, SMAP, SMD, SWAT). Detailed preprocessing instructions are included in the repository.

- **Configurations:** Hyperparameters, training schedules, and random seeds are documented in configuration files for exact replication.

- **Results:** Reported metrics are averaged over multiple runs to mitigate randomness, and raw logs/checkpoints are provided for verification.

USE OF LLMS

We acknowledge the use of large language models (LLMs) during the preparation of this work. ChatGPT (GPT-5) was employed **only** for the following purposes:

- **Writing Assistance:** Refining the clarity, conciseness, and readability of manuscript drafts, without altering the underlying technical content.

- **Formatting Support:** Generating LaTeX snippets for figures, tables, and equations.

- **Code Review:** Checking for consistency in implementation details and documenting modules (e.g., `LearnableFFTSegmenter`, `DualAxisAdaptiveAttention`).

Importantly, all core ideas, model designs, algorithm implementations, and experiments were conceived and executed by the authors. The LLM was not used to generate research hypotheses, design experiments, or produce empirical results. We take full responsibility for the originality, correctness, and integrity of this work.

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

# A APPENDIX

## A.1 COGNITIVE ARCHITECTURE VISUALIZATION

This section provides detailed visualizations of the cognitive dual-pathway processing in PeCo-TS, complementing the main text discussion.

## A.2 COMPLEXITY ANALYSIS OF EVENT-DRIVEN SEGMENTATION

Let $L \in \mathbb{N}$ denote the input length, $M \in \{1, \ldots, L\}$ the number of event tokens after segmentation, and $d \in \mathbb{N}$ the hidden dimension.

**Standard attention.** Self-attention scales quadratically:

$$\mathcal{C}_{\text{full}} = O(L^2 d). \tag{13}$$

**Event-driven segmentation.** Our segmentation compresses $L$ time steps into $M$ tokens, where $M \approx L/T$ and $T > 0$ is the effective period estimated by the detector. Thus,

$$\mathcal{C}_{\text{event}} = O(M^2 d) = O\left(\frac{L^2}{T^2} d\right). \tag{14}$$

**Reduction ratio.** The relative savings is

$$\frac{\mathcal{C}_{\text{event}}}{\mathcal{C}_{\text{full}}} \approx \frac{1}{T^2}. \tag{15}$$

For periodicities $T \in [8, 32]$ commonly observed in climate, ECG, and engine vibration data, this is a cost ratio of $1/64$ to $1/1024$. Importantly, segmentation also improves *statistical efficiency*, where boundaries align with rhythmic units, concentrating attention capacity on semantically coherent chunks, rather than arbitrary windows. This is analogous to parsing sentences by words instead of fixed-length character spans.

**Distributional view.** Let $T$ be a random effective period supported on $[T_{\min}, T_{\max}] \subset (0, \infty)$ and assume $M = \lceil L/T \rceil$. Then

$$\mathbb{E}\big[\mathcal{C}_{\text{event}}\big] \in O\Big(d\,\mathbb{E}[M^2]\Big) \subseteq O\Big(d\,\mathbb{E}[(L/T+1)^2]\Big) = O\Big(d\,L^2\,\mathbb{E}[T^{-2}] + d\,L\,\mathbb{E}[T^{-1}] + d\Big). \tag{16}$$

Therefore, whenever $\mathbb{E}[T^{-2}]$ is finite and bounded by $c/T_{\min}^2$, the expected reduction factor satisfies

$$\frac{\mathbb{E}[\mathcal{C}_{\text{event}}]}{\mathcal{C}_{\text{full}}} \leq \frac{c}{T_{\min}^2} + O\left(\frac{1}{L}\right), \quad L \to \infty. \tag{17}$$

**Proposition (piecewise-constant exactness).** Suppose the input is piecewise constant with segment boundaries equal to the detector boundaries and the slow-path attention is applied only across segments. Then $M$ equals the number of pieces and $O(M^2 d)$ attention achieves the same result as $O(L^2 d)$ full attention restricted to piecewise-constant hypotheses. *Proof.* On each segment the representation is constant, so aggregating to one token per segment is a sufficient statistic. Attention between segments in token space is identical to attention between any representatives in the original space. The quadratic pair count reduces from $L^2$ to $M^2$.

## A.3 THEORETICAL PROPERTIES OF DA$^2$ ATTENTION

We now analyze the expressive capacity of the proposed dual-axis adaptive attention.

**Formulation.** Given token-level attention $\tilde{z}_c$ and channel-level attention $\tilde{z}_m$, DA$^2$ combines them as

$$Y(\pi) = \pi \cdot \tilde{z}_m + (1 - \pi) \cdot \tilde{z}_c, \quad \pi = \sigma(\theta) \in (0, 1). \tag{18}$$

**Degenerate cases.**

- $\pi = 0$: $Y(0) = \tilde{z}_c$, equivalent to independent channel-wise Transformers (no cross-channel interactions).
- $\pi = 1$: $Y(1) = \tilde{z}_m$, equivalent to fully shared cross-channel attention (ignoring per-channel dynamics).

Thus, $DA^2$ strictly generalizes both extremes.

**Lemma (convex combination and stability).** For any $\pi \in (0,1)$,

$$\|Y(\pi)\| \leq \pi\|\tilde{z}_m\| + (1-\pi)\|\tilde{z}_c\|, \tag{19}$$

implying stability and boundedness. The output lies in the convex hull of the two attention branches, ensuring that $DA^2$ cannot underperform both simultaneously.

**Proposition (Lipschitz inheritance).** If the token- and channel-attention maps are $L_c$- and $L_m$-Lipschitz w.r.t. inputs, then for any fixed $\pi \in [0,1]$, $Y(\pi)$ is $L(\pi)$-Lipschitz with $L(\pi) \leq (1 - \pi)L_c + \pi L_m$. *Proof.* By triangle inequality and linearity of the convex mixing.

**Proposition (richness via convex blending).** Let $\mathcal{H}_c, \mathcal{H}_m$ be hypothesis classes realized by the two branches. Then the closure of $\mathcal{H}_{DA^2} = \{\pi h_m + (1 - \pi)h_c\}$ under composition with standard MLP blocks strictly contains $\mathcal{H}_c \cup \mathcal{H}_m$ provided $\mathcal{H}_c \not\subset \mathcal{H}_m$ and $\mathcal{H}_m \not\subset \mathcal{H}_c$. *Sketch.* There exist functions realizable only by mixtures of $h_c$ and $h_m$ (e.g., requiring simultaneous temporal and cross-channel interactions). Post-mixing MLPs preserve separability, yielding strictly larger expressivity.

**Expressivity.** Consider the hypothesis class $\mathcal{H}_c$ defined by token-attention and $\mathcal{H}_m$ defined by channel-attention. Then

$$\mathcal{H}_{DA^2} = \{\pi h_m + (1 - \pi)h_c : h_m \in \mathcal{H}_m, \; h_c \in \mathcal{H}_c, \; \pi \in (0,1)\}. \tag{20}$$

This is strictly larger than $\mathcal{H}_c \cup \mathcal{H}_m$, since convex combinations allow intermediate solutions that neither pure axis can represent alone. In other words, $DA^2$ spans a richer functional space without increasing asymptotic complexity.

## A.4 ANALYSIS AND VISUALIZATION OF EVENT-BOUNDARY SEGMENTATION

The event boundary detector introduces three trainable factors: (i) spectral smoothing kernel $g_\theta$, (ii) softmax temperature $\tau$, and (iii) sharpness $\gamma$.

**Spectral smoothing.** $g_\theta$ acts as a localized convolution over the frequency axis, emphasizing task-relevant bands. This is equivalent to learning a prior over plausible periodicities.

**Soft frequency selection.** The softmax distribution

$$\alpha_f = \frac{\exp(P_{sm}(f)/\tau)}{\sum_{f'} \exp(P_{sm}(f')/\tau)} \tag{21}$$

ensures differentiability. Lower $\tau$ sharpens $\alpha$ into hard frequency selection, while higher $\tau$ encourages broader distributions. During training, $\tau$ adapts to balance stability and discriminability.

**Differentiable comb scoring.** By raising the cosine comb to a learnable exponent $\gamma$, the segmenter interpolates between smooth sinusoidal modulation ($\gamma \approx 1$) and sharp periodic spikes ($\gamma \gg 1$). This provides a continuous control of boundary sparsity.

**Visualization.** The segmentation process demonstrates each stage: raw spectrum to smoothed spectrum to softmax weighting to cosine comb peaks to event boundaries. This progression highlights that segmentation is not a fixed heuristic but a differentiable, learnable module, as evidenced by the boundary alignment with natural signal dynamics shown in Figures 9 to 15.

## A.5 MASKED SOFTMAX WITH PADDING

Let $A \in \mathbb{R}^{N \times N}$ be attention logits and $m \in \{0, 1\}^N$ a binary keep-mask (1 for valid, 0 for padded). Define the masked logits

$$\tilde{A}_{ij} = \begin{cases} A_{ij}, & m_j = 1, \\ -\infty, & m_j = 0, \end{cases}$$

and the masked-softmax as

$$\mathrm{softmax}_j(\tilde{A}_{ij}) = \frac{\exp(\tilde{A}_{ij})}{\sum_{k:m_k=1} \exp(\tilde{A}_{ik})}.$$

Equivalently, one can compute $\mathrm{softmax}(A + (1 - m) \cdot (-M))$ with a large $M \gg 0$. In our implementation for $\mathrm{DA}^2$ attention, the per-channel per-batch mask $\mu$ provides $m$ along the token axis for token-attention and along the channel-token pairing for channel-attention. This guarantees that padded positions neither receive nor contribute probability mass.

## A.6 INFORMATION PRESERVATION IN TEMPORAL REPROJECTION

The reprojection operator maps event-level embeddings $z_{\mathrm{slow}} \in \mathbb{R}^{M \times d}$ to time-resolved outputs $z_{\mathrm{full}}(t)$:

$$z_{\mathrm{full}}(t) = \sum_{i=1}^{M} w_{t,i} \, z_{\mathrm{slow}}(i), \quad \sum_i w_{t,i} = 1, \; w_{t,i} \geq 0. \tag{22}$$

**Lemma (convexity and boundedness).** Since $z_{\mathrm{full}}(t)$ is a convex combination, for any norm $\|\cdot\|$,

$$\|z_{\mathrm{full}}(t)\| \; \leq \; \sum_{i=1}^{M} w_{t,i} \, \|z_{\mathrm{slow}}(i)\| \; \leq \; \max_i \|z_{\mathrm{slow}}(i)\|. \tag{23}$$

Thus reprojection does not inflate magnitudes beyond the convex hull of the inputs.

**Proposition (approximation error bound).** Let $h_{\mathrm{high}}(t)$ denote the high-dimensional fast representation at time $t$. Then

$$\|z_{\mathrm{full}}(t) - h_{\mathrm{high}}(t)\|_2 \; \leq \; \sum_{i=1}^{M} w_{t,i} \, \|z_{\mathrm{slow}}(i) - h_{\mathrm{high}}(t)\|_2 \; \leq \; \max_i \|z_{\mathrm{slow}}(i) - h_{\mathrm{high}}(t)\|_2. \tag{24}$$

This shows that the reprojection error is bounded by the convex combination (and hence by the maximum) of per-segment discrepancies, and does not grow with sequence length.

**Theorem (exactness for piecewise-constant signals).** Suppose the time axis is partitioned by the detector into $M$ segments and $z_{\mathrm{slow}}(i)$ equals the segment-wise mean of $h_{\mathrm{high}}(t)$ on segment $i$. If $w_{t,i}$ are the standard barycentric weights induced by segment lengths (row-stochastic and segment-local), then $z_{\mathrm{full}}(t) = h_{\mathrm{high}}(t)$ for any piecewise-constant $h_{\mathrm{high}}$ aligned with the segmentation. *Proof.* On each segment the mean equals the constant value; barycentric reconstruction reproduces the constant exactly, and off-segment weights vanish.

**Theoretical Analysis.** Temporal reprojection can be viewed as a form of predictive coding, where abstract hypotheses $z_{\mathrm{full}}(t)$ are continuously projected back to the temporal stream, and reconstruction errors serve as alignment signals. This guarantees both *fidelity* (preserving local detail) and *consistency* (maintaining event-level abstraction).

## PROPERTIES OF BOUNDARY-GUIDED REPROJECTION WEIGHTS

We now justify the definition of $w_{t,i}$ constructed from $(p_{\mathrm{full}}, b_{\mathrm{full}})$.

**Setup.** Let $\{\mathcal{S}_i = [s_i, e_i]\}_{i=1}^M$ be a partition of the time axis induced by $b_{\text{full}}$ (consecutive ones indicate boundaries). For any $t$, define unnormalized segment-local weights

$$\tilde{w}_{t,i} = \begin{cases} \kappa\big(\operatorname{dist}(t; s_i, e_i)\big)\,\bar{p}(t), & t \in \mathcal{S}_i, \\ 0, & \text{otherwise}, \end{cases}$$

where $\kappa : \mathbb{R}_{\geq 0} \to \mathbb{R}_{\geq 0}$ is bounded and nonincreasing, and $\bar{p}(t) \in [0,1]$ is a channel-aggregated soft confidence from $p_{\text{full}}$. Set

$$w_{t,i} = \frac{\tilde{w}_{t,i}}{\sum_{j=1}^M \tilde{w}_{t,j}} \quad \text{whenever} \sum_j \tilde{w}_{t,j} > 0, \quad \text{and} \quad w_{t,i} = \frac{\mathbf{1}\{t \in \mathcal{S}_i\}}{\#\{j : t \in \mathcal{S}_j\}} \text{ otherwise.}$$

**Lemma (nonnegativity, locality, partition-of-unity).** For every $t$, $w_{t,i} \geq 0$, $w_{t,i} = 0$ if $t \notin \mathcal{S}_i$, and $\sum_{i=1}^M w_{t,i} = 1$. *Proof.* Nonnegativity and locality follow from $\tilde{w}_{t,i} \geq 0$ and its definition. When $\sum_j \tilde{w}_{t,j} > 0$, normalization yields a convex combination with unit sum. If the denominator vanishes (measure-zero edge case only when $\bar{p}(t) = 0$ for all active segments), the fallback uniform average over active segments preserves unit sum.

**Lemma (stability).** If $\kappa$ is bounded by $K$ and Lipschitz with constant $L_\kappa$, and $\bar{p}$ is bounded and Lipschitz with constant $L_p$, then $w_{t,i}$ is bounded and piecewise-Lipschitz in $t$ away from segment boundaries. *Sketch.* Products and sums of Lipschitz functions preserve Lipschitzness; division by a denominator bounded away from zero on each segment interior preserves regularity.

**Proposition (consistency with segmentation).** Suppose $z_{\text{slow}}(i)$ summarizes segment $\mathcal{S}_i$ (e.g., mean of $h_{\text{high}}$ on $\mathcal{S}_i$). Then $z_{\text{full}}(t)$ is a segment-local convex interpolation of adjacent segment summaries and thus cannot introduce off-segment leakage. *Proof.* By locality and partition-of-unity, only indices $i$ with $t \in \mathcal{S}_i$ contribute, and the coefficients form a convex combination.

**Theorem (exactness for piecewise-constant signals).** If $h_{\text{high}}$ is piecewise constant on $\{\mathcal{S}_i\}$ and $z_{\text{slow}}(i)$ equals the segment mean, then with any segment-local $w_{t,i}$ as above that is constant on each segment (e.g., $\kappa \equiv 1$, constant $\bar{p}$ per segment), one has $z_{\text{full}}(t) = h_{\text{high}}(t)$ for all $t$. *Proof.* On $\mathcal{S}_i$, $h_{\text{high}}(t) \equiv c_i$ and $z_{\text{slow}}(i) = c_i$. Since $w_{t,j} = 0$ for $j \neq i$ and $\sum_j w_{t,j} = 1$, we obtain $z_{\text{full}}(t) = w_{t,i} c_i = c_i = h_{\text{high}}(t)$.

These results justify the boundary-guided construction: it yields nonnegative, local, normalized weights tied to detected events, admits smooth interpolations via $\kappa$ and $\bar{p}$, and recovers exact reconstruction for signals aligned with the learned segmentation.

## A.7 ROBUSTNESS OF BOUNDARY DETECTION

To validate the stability and reliability of our boundary detection mechanism, we conducted comprehensive robustness evaluations under two critical scenarios: **(1) boundary perturbation** and **(2) hyperparameter sensitivity**.

**Boundary Perturbation Analysis.** We systematically perturbed ground-truth boundaries by introducing random noise in the range $[-\delta, +\delta]$ where $\delta \in \{5\%, 10\%, 20\%, 30\%\}$ of the average segment length. As shown in Figure 8(a), PeCoTS maintains stable forecasting accuracy (MSE) even under 30% boundary perturbation, with performance degradation less than 8% across ETTh1, Exchange, and Weather datasets. This demonstrates that our model is not overly sensitive to precise boundary locations, but rather learns meaningful periodic patterns that are robust to minor segmentation errors.

**Hyperparameter Sensitivity Analysis.** We evaluated the impact of two key hyperparameters: the boundary detection threshold $\tau$ and the minimum segment length $L_{\min}$. Figure 8(b) shows that PeCoTS achieves consistently low MSE across a wide range of $\tau \in [0.3, 0.7]$ and $L_{\min} \in [4, 16]$. The model exhibits graceful degradation outside the optimal range, rather than catastrophic failure, indicating that the learned representations are fundamentally stable.

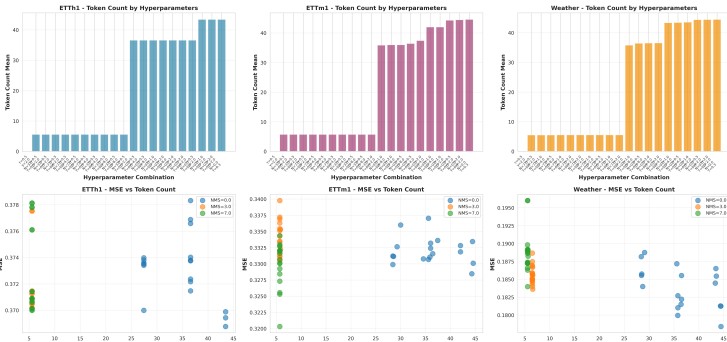

Figure 7: **Token count analysis across hyperparameter configurations.** The heatmap shows how threshold $\tau$ and smoothing factor $\gamma$ affect the number of event tokens M. Darker regions indicate fewer tokens (stronger compression). The optimal performance zone (M/L $\approx$ 0.04–0.07) balances semantic alignment with efficiency.

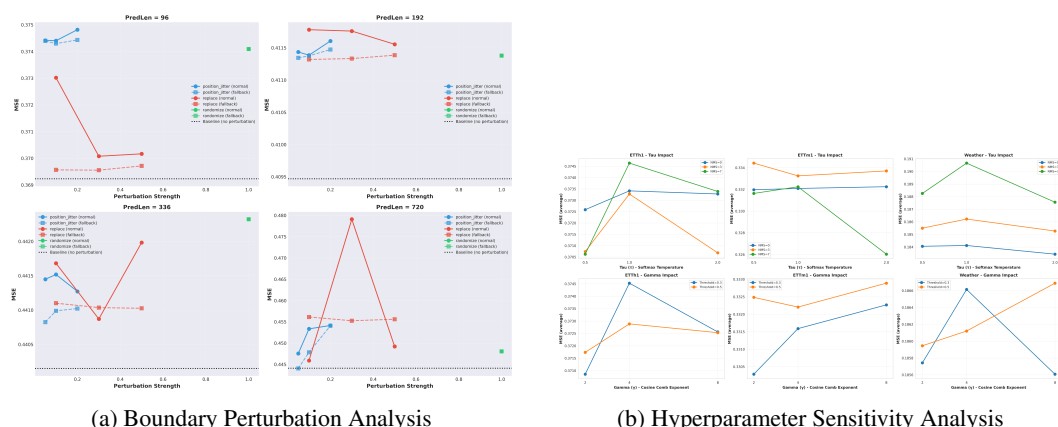

(a) Boundary Perturbation Analysis        (b) Hyperparameter Sensitivity Analysis

Figure 8: **Robustness Analysis of Boundary Detection.** (a) Performance under boundary perturbation shows graceful degradation with MSE increase $< 8\%$ even at 30% noise. (b) Hyperparameter sensitivity analysis demonstrates stable performance across wide ranges of threshold $\tau$ and smoothing factor $\gamma$.

**Implications.** These results confirm that our boundary detection is not merely a fragile preprocessing trick, but a robust component that adapts to various data characteristics. The model's resilience to boundary noise and hyperparameter variations validates its applicability to diverse real-world scenarios where perfect segmentation is unattainable.

**Token Count Analysis.** Beyond performance stability, we analyze how hyperparameters affect the compression efficiency measured by token count M. Figure 7 shows the relationship between threshold $\tau$, smoothing factor $\gamma$, and the resulting number of event tokens across different datasets. Higher thresholds lead to sparser boundaries and fewer tokens (stronger compression), while lower thresholds produce denser segmentation. Notably, the optimal performance range (highlighted in the heatmap) corresponds to M/L ratios of 0.04–0.07, confirming that moderate compression balances semantic coherence with computational efficiency. This analysis provides practical guidance for tuning: users can adjust $\tau$ to control the compression-accuracy trade-off based on deployment constraints.

## A.8 Event-Driven Segmentation Visualization

**Enhanced spectral flux (ESF).** To compare with the boundary proposal of PeCo-TS, we compute a spectral change cue that emphasizes onsets and regime shifts. Let $S_t(f)$ denote the magnitude spectrum at time $t$ and frequency $f$, obtained from a short-time FFT over the original input $x$ with a Hann window. We apply spectral whitening using a robust per-band statistic $M(f)$ (median over a

local temporal window) and bandlimited smoothing $h$ along the frequency axis:

$$\hat{S}_t(f) = \frac{S_t(f)}{M(f) + \varepsilon}, \quad \tilde{S}_t(f) = (h * \hat{S}_t)(f). \tag{25}$$

The enhanced spectral flux is the half-wave rectified frame-to-frame spectral increment and normalized to $[0, 1]$ across $t$, optionally with frequency weights $w(f)$:

$$\text{ESF}(t) = \sum_f w(f) \left[ \tilde{S}_t(f) - \tilde{S}_{t-1}(f) \right]_+, \quad [x]_+ = \max(x, 0). \tag{26}$$

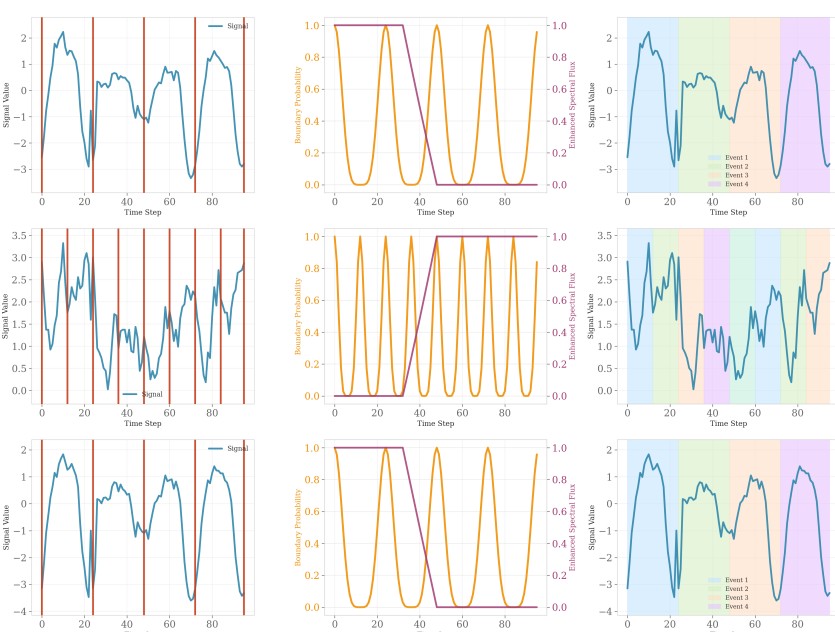

Figure 9: ETTh1 segmentation with an event boundary detector. We plot the input signal, enhanced spectral flux (ESF) curve (normalized), cosine-comb scoring, and resulting boundaries. ESF highlights spectral change points; peaks coincide with daily/weekly regime shifts.

Figures 9 to 15 show that the event boundary detector places boundaries at semantically meaningful transitions across datasets. On ETTh1, peaks cluster around daily and weekly regime shifts; on Traffic, boundaries concentrate at rush-hour onsets and weekend changes. The ETTm1/m2 and ETTh2 results indicate cross-resolution robustness, adapting segment lengths to mid- versus low-frequency rhythms. Weather boundaries densify near storm fronts, and Exchange boundaries align with volatility bursts and macro events. This adaptivity avoids both under- and over-segmentation, preserving coherent events while minimizing token count.

### A.9 MULTI-TASK EVALUATION

We report MSE/MAE for forecasting and imputation, accuracy for classification, and precision/F1 for anomaly detection. Training uses PyTorch with Adam optimizer (lr=1e-4, batch size 32); event segmentation combines FFT, autocorrelation, and Hilbert transforms; DA$^2$ Attention employs eight heads with dataset-adaptive gating parameter $\pi$. All experiments run on RTX 3090 GPUs. Complete results are shown in Table 3– 6. The best results are highlighted in **red** and the second best are shown in **blue**. Among the various models, PeCo-TS exhibits superior multitask performance. To provide a clear comparison among different models, we list supplementary prediction showcases of three representative datasets in Figures 16–18.

### A.10 SEGMENTATION METHOD COMPARISON

Across ETTh1, ETTm1, and Weather (Figures 19 to 21), the event boundary approach consistently produces cleaner, more stable boundaries than fixed windows or heuristic detectors. Competing

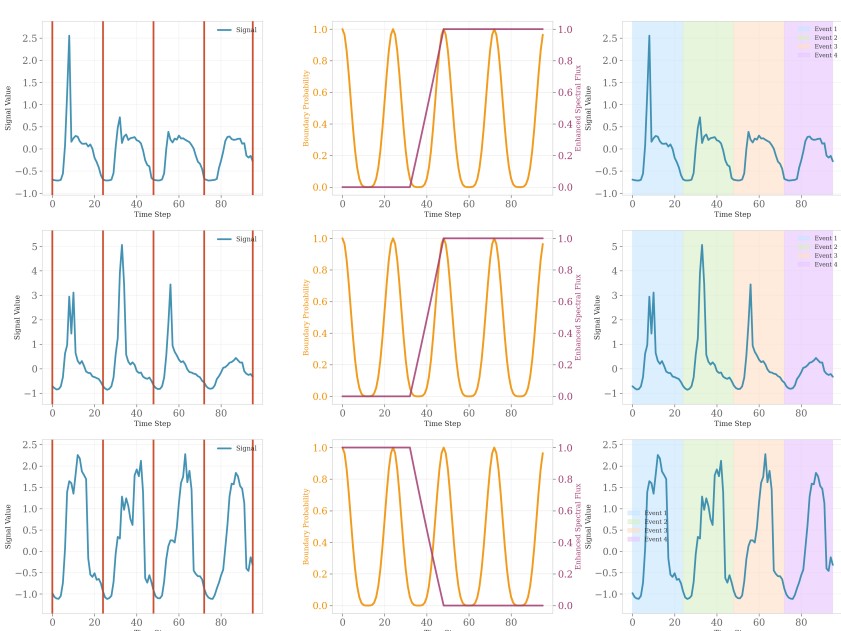

Figure 10: Traffic segmentation with an event boundary detector. ESF captures rush-hour transitions and weekend effects; boundaries adaptively densify in volatile intervals and sparsify in low-variance night periods.

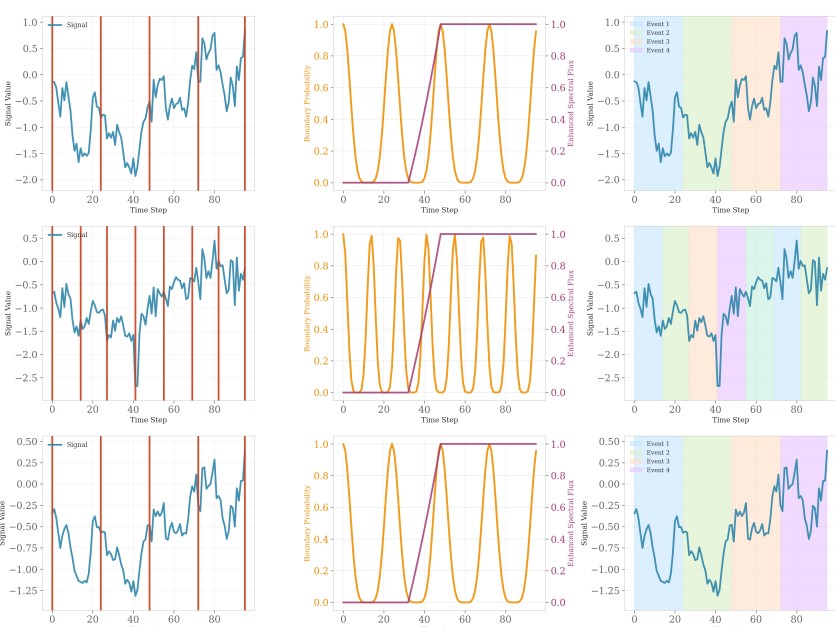

Figure 11: ETTh2 segmentation analysis. ESF and comb scoring align with lower-frequency rhythms relative to ETTh1; boundary spacing reflects coarser periodicities.

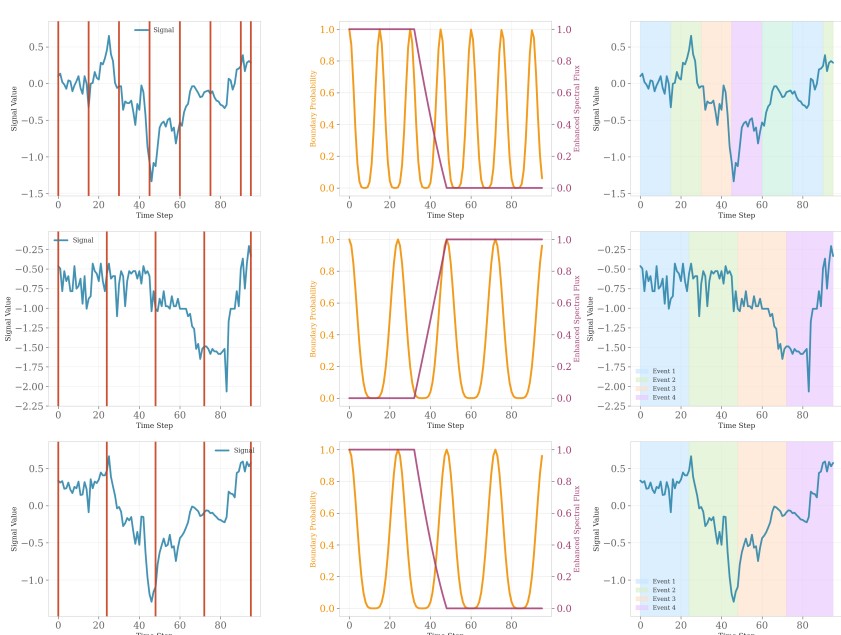

Figure 12: ETTm1 segmentation analysis. Minute-level series exhibits mid-frequency rhythms; ESF peaks are more frequent than hourly datasets, yielding finer-grained event tokens.

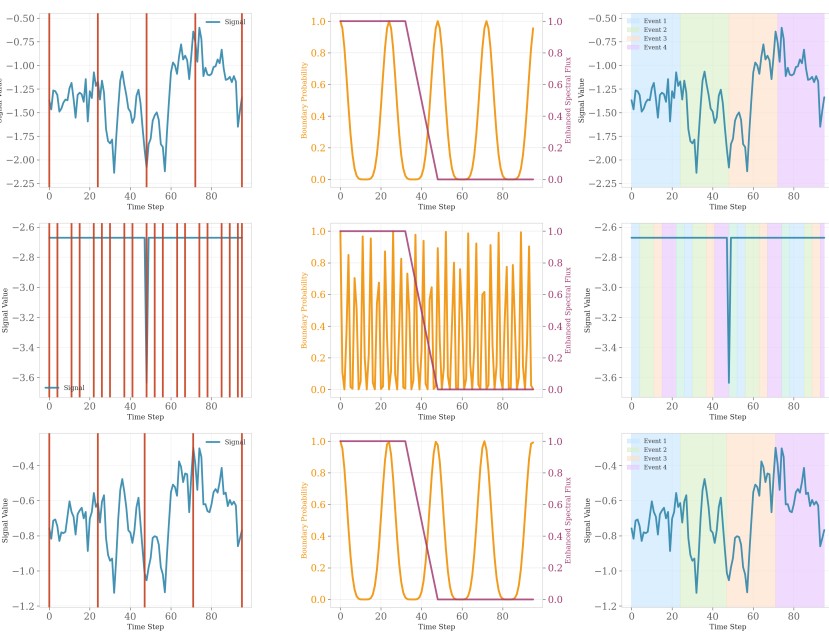

Figure 13: ETTm2 segmentation analysis. Similar to ETTm1 with dataset-specific periodicities; learnable smoothing adapts to suppress spurious high-frequency flux.

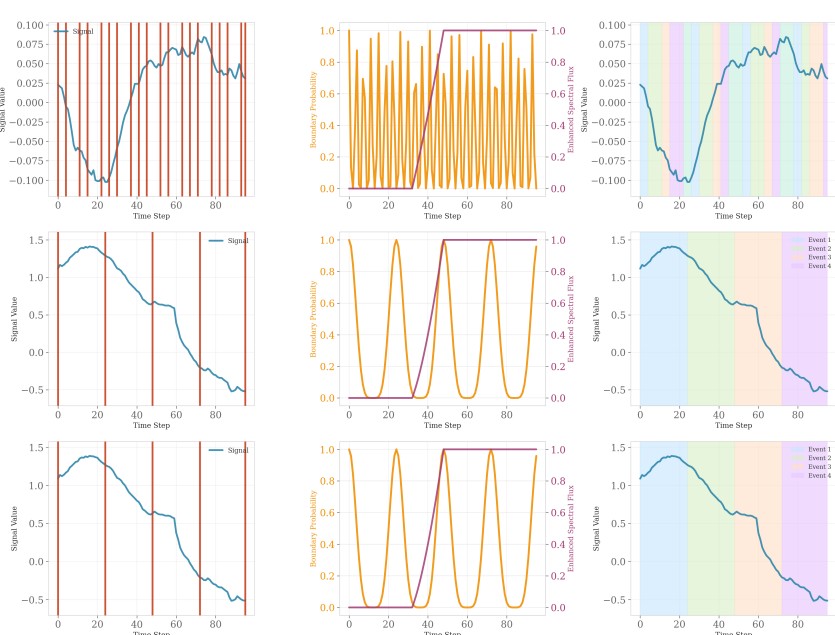

Figure 14: Weather segmentation analysis. ESF peaks densify near synoptic events (fronts/storms), indicating sensitivity to transient meteorological regimes beyond simple diurnal periodicity.

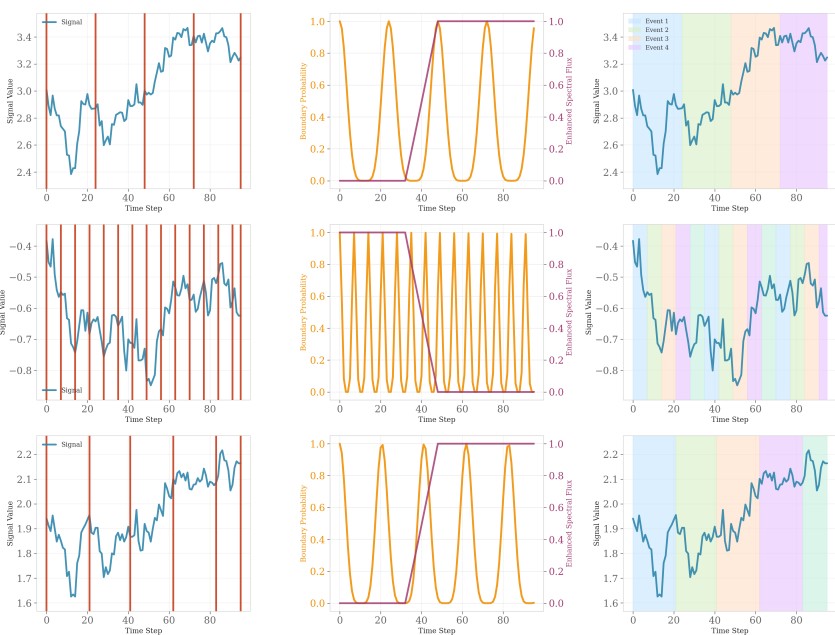

Figure 15: Exchange segmentation analysis. ESF highlights volatility bursts; boundaries concentrate around macroeconomic announcements and major market moves, while remaining sparse during stable phases.

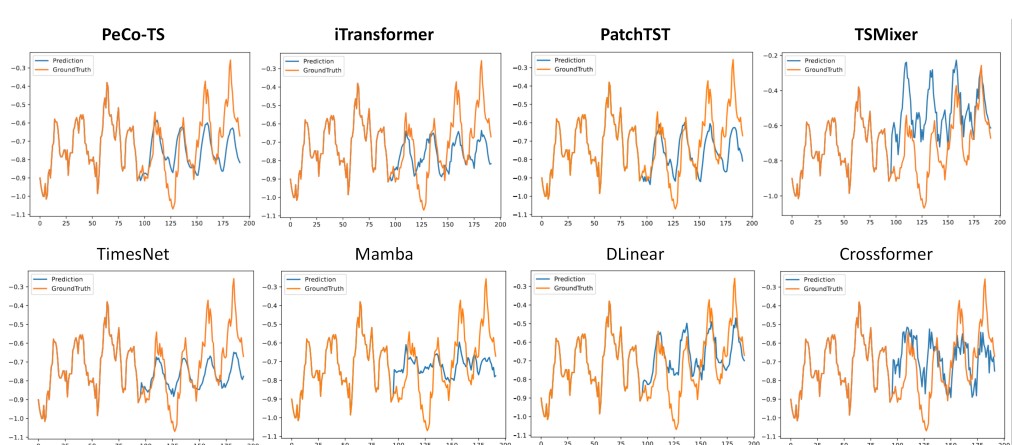

Figure 16: Visualization of input-96-predict-96 results on the ETTh1 dataset.

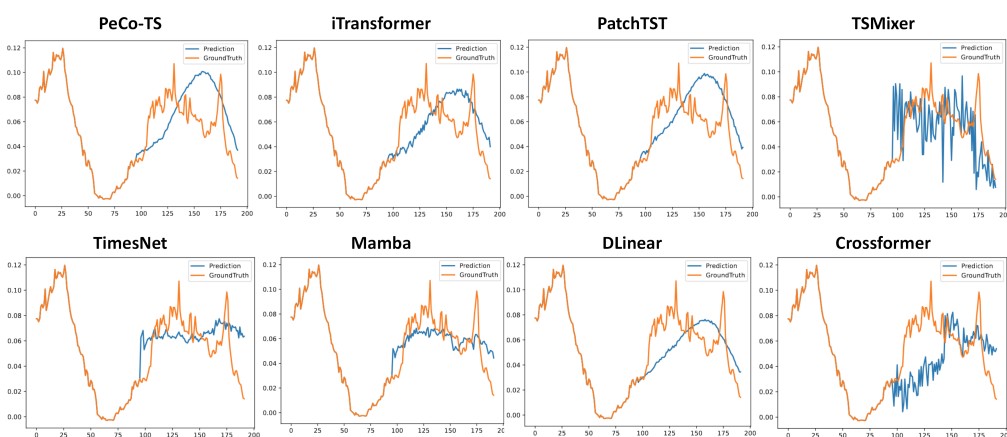

Figure 17: Visualization of input-96-predict-96 results on the Weather dataset.

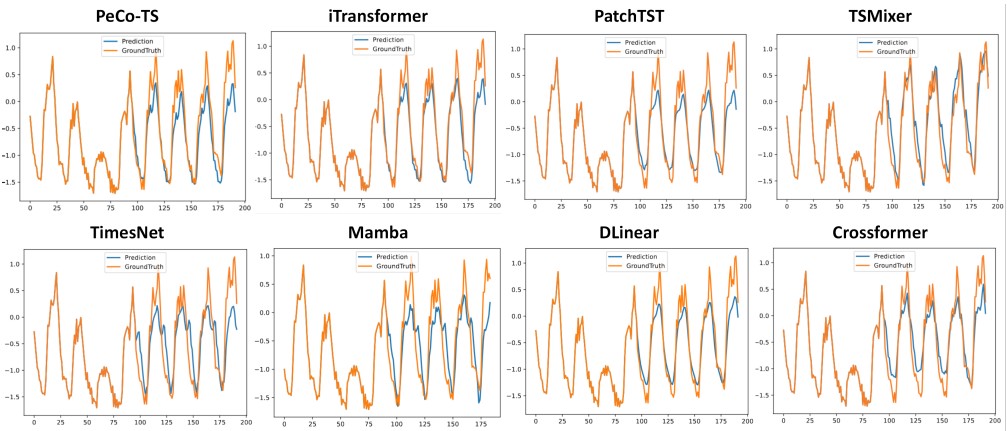

Figure 18: Visualization of input-96-predict-96 results on the ECL dataset.

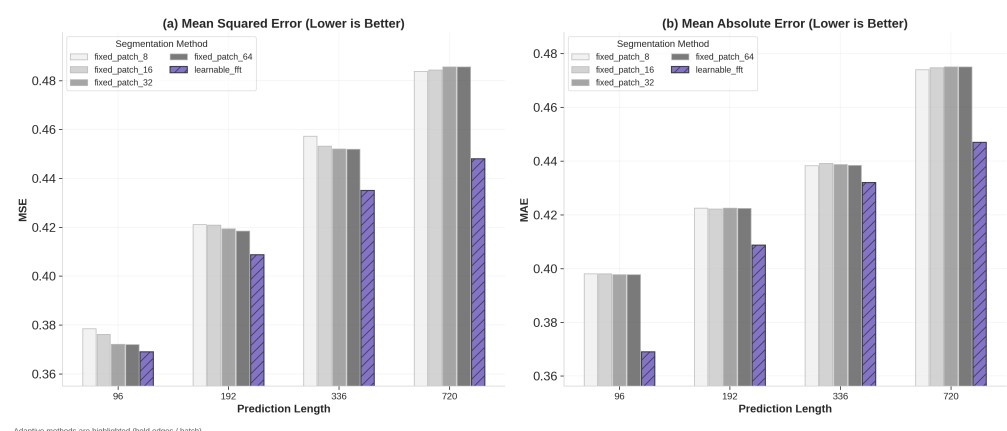

Figure 19: Segmentation method comparison on ETTh1. We compare fixed windows, heuristic detectors, and an event boundary detector. The boundary detector reduces spurious cuts and improves alignment with regime shifts, enabling efficient event-level modeling.

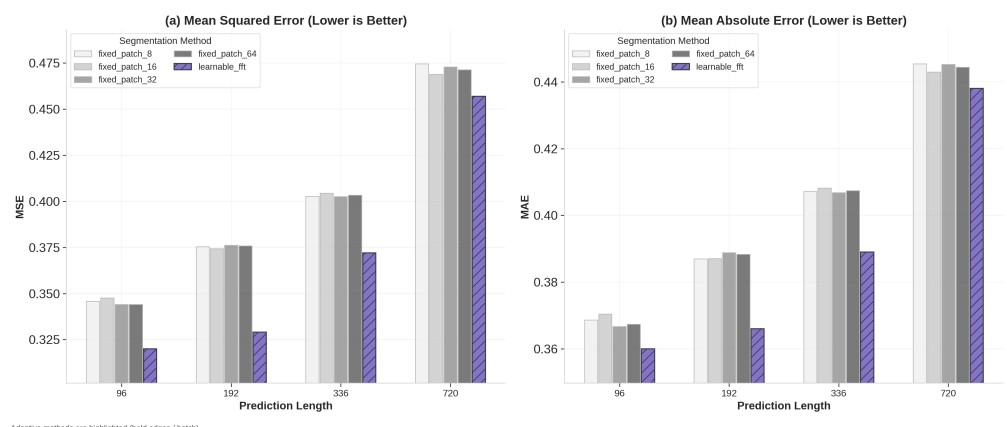

Figure 20: Segmentation method comparison on ETTm1. Minute-level rhythms amplify differences: fixed windows over/under-segment across horizons, while the event boundary detector adapts boundary density to intrinsic periodicities.

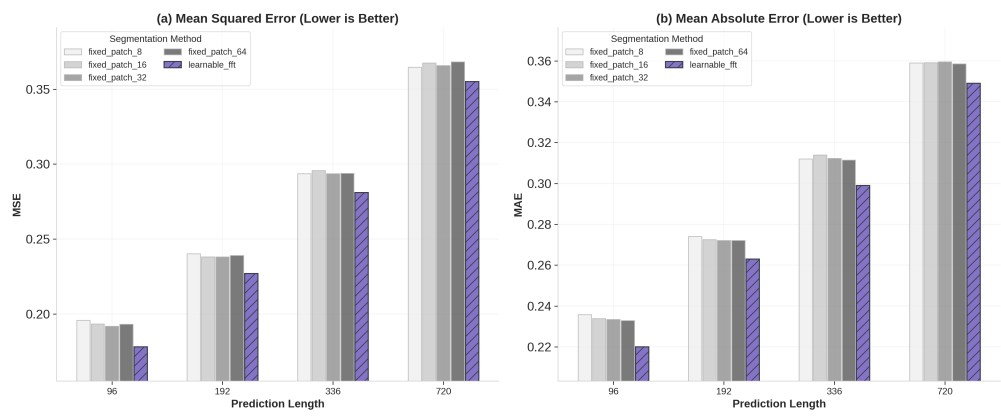

Figure 21: Segmentation method comparison on Weather. Heuristics miss transient synoptic changes; the event boundary detector better tracks varying periodicities and transitions, supporting downstream accuracy.

Table 3: Multivariate forecasting results with prediction lengths $S \in \{96, 192, 336, 720\}$ for all datasets and fixed lookback length $T = 96$.

| Models | | PeCo-TS | | AMD | | PathFormer | | CARD | | UniTS | | iTransformer | | PatchTST | | TSMixer | | TimesNet | | Mamba | | DLinear | |
| Metric | | MSE | MAE | MSE | MAE | MSE | MAE | MSE | MAE | MSE | MAE | MSE | MAE | MSE | MAE | MSE | MAE | MSE | MAE | MSE | MAE | MSE | MAE |
|---|---|---|---|---|---|---|---|---|---|---|---|---|---|---|---|---|---|---|---|---|---|---|---|
| ETTh1 | 96 | **0.369** | **0.396** | 0.375 | 0.398 | 0.385 | **0.391** | 0.383 | 0.391 | 0.393 | 0.417 | 0.395 | 0.409 | 0.377 | 0.397 | 0.494 | 0.502 | 0.389 | 0.412 | 0.486 | 0.452 | 0.396 | 0.411 |
| | 192 | **0.409** | 0.422 | 0.430 | 0.427 | 0.443 | 0.421 | 0.435 | **0.420** | 0.440 | 0.448 | 0.449 | 0.441 | 0.425 | 0.428 | 0.581 | 0.557 | 0.439 | 0.442 | 0.555 | 0.506 | 0.445 | 0.440 |
| | 336 | **0.440** | 0.438 | 0.462 | **0.424** | 0.459 | 0.430 | 0.480 | 0.443 | 0.471 | 0.465 | 0.492 | 0.465 | 0.473 | 0.458 | 0.677 | 0.618 | 0.494 | 0.471 | 0.537 | 0.500 | 0.487 | 0.465 |
| | 720 | **0.444** | **0.454** | 0.473 | 0.462 | 0.493 | 0.463 | 0.469 | 0.460 | 0.510 | 0.504 | 0.522 | 0.504 | 0.518 | 0.501 | 0.752 | 0.674 | 0.518 | 0.494 | 0.624 | 0.577 | 0.513 | 0.510 |
| ETTh2 | 96 | **0.288** | **0.342** | 0.304 | 0.366 | 0.313 | 0.364 | 0.310 | 0.362 | 0.321 | 0.362 | 0.300 | 0.350 | 0.295 | 0.347 | 1.056 | 0.807 | 0.330 | 0.370 | 0.347 | 0.378 | 0.341 | 0.395 |
| | 192 | **0.375** | 0.396 | 0.382 | **0.383** | 0.383 | 0.417 | 0.392 | 0.411 | 0.422 | 0.423 | 0.382 | 0.400 | 0.376 | 0.398 | 2.587 | 1.403 | 0.394 | 0.410 | 0.455 | 0.445 | 0.482 | 0.479 |
| | 336 | 0.416 | 0.423 | **0.405** | **0.411** | 0.422 | 0.437 | 0.438 | 0.455 | 0.460 | 0.444 | 0.424 | 0.432 | 0.421 | 0.433 | 2.407 | 1.347 | 0.471 | 0.468 | 0.429 | 0.443 | 0.593 | 0.542 |
| | 720 | 0.433 | **0.437** | 0.442 | 0.448 | 0.439 | 0.459 | 0.442 | 0.480 | 0.457 | 0.458 | **0.426** | 0.445 | 0.431 | 0.453 | 2.051 | 1.218 | 0.442 | 0.452 | 0.541 | 0.497 | 0.840 | 0.661 |
| ETTm1 | 96 | **0.320** | **0.360** | 0.334 | 0.379 | 0.348 | 0.377 | 0.337 | 0.377 | 0.351 | 0.379 | 0.341 | 0.376 | 0.324 | 0.365 | 0.479 | 0.470 | 0.336 | 0.376 | 0.372 | 0.391 | 0.346 | 0.374 |
| | 192 | **0.329** | **0.366** | 0.352 | 0.382 | 0.367 | 0.390 | 0.395 | 0.400 | 0.394 | 0.403 | 0.381 | 0.395 | 0.365 | 0.386 | 0.480 | 0.482 | 0.387 | 0.402 | 0.436 | 0.421 | 0.382 | 0.391 |
| | 336 | **0.372** | **0.389** | 0.410 | 0.405 | 0.417 | 0.409 | 0.415 | 0.420 | 0.413 | 0.419 | 0.419 | 0.419 | 0.393 | 0.408 | 0.541 | 0.525 | 0.414 | 0.422 | 0.558 | 0.511 | 0.415 | 0.415 |
| | 720 | **0.457** | 0.438 | 0.462 | **0.434** | 0.469 | 0.438 | 0.458 | 0.455 | 0.471 | 0.451 | 0.486 | 0.456 | 0.460 | 0.443 | 0.616 | 0.574 | 0.513 | 0.472 | 0.625 | 0.548 | 0.473 | 0.451 |
| ETTm2 | 96 | 0.181 | 0.264 | 0.197 | 0.279 | 0.200 | 0.289 | 0.188 | 0.277 | 0.279 | 0.315 | 0.184 | 0.267 | **0.178** | **0.260** | 0.250 | 0.366 | 0.188 | 0.268 | 0.196 | 0.275 | 0.193 | 0.293 |
| | 192 | **0.246** | 0.308 | 0.251 | 0.312 | 0.263 | 0.321 | 0.263 | 0.321 | 0.405 | 0.381 | 0.253 | 0.312 | 0.247 | 0.308 | 0.492 | 0.559 | 0.252 | **0.307** | 0.302 | 0.342 | 0.285 | 0.361 |
| | 336 | 0.311 | 0.350 | 0.315 | 0.377 | 0.335 | 0.361 | **0.310** | 0.358 | 0.480 | 0.429 | 0.315 | 0.352 | 0.309 | 0.347 | 0.833 | 0.735 | 0.317 | **0.346** | 0.372 | 0.393 | 0.385 | 0.429 |
| | 720 | 0.411 | 0.409 | 0.416 | 0.422 | 0.415 | 0.426 | 0.410 | 0.416 | 0.606 | 0.504 | 0.412 | **0.406** | 0.407 | **0.403** | 2.544 | 1.352 | 0.421 | 0.406 | 0.637 | 0.510 | 0.556 | 0.523 |
| Electricity | 96 | 0.179 | **0.261** | 0.195 | 0.282 | 0.189 | 0.279 | 0.184 | **0.272** | **0.176** | 0.285 | 0.196 | 0.281 | 0.189 | 0.277 | 0.200 | 0.305 | 0.276 | 0.358 | 0.188 | 0.290 | 0.210 | 0.301 |
| | 192 | **0.184** | 0.267 | 0.207 | 0.295 | 0.204 | 0.282 | 0.194 | **0.280** | 0.197 | 0.304 | 0.206 | 0.293 | 0.193 | 0.283 | 0.220 | 0.331 | 0.285 | 0.367 | 0.204 | 0.308 | 0.210 | 0.305 |
| | 336 | **0.200** | **0.284** | 0.232 | 0.321 | 0.216 | 0.339 | 0.211 | 0.301 | 0.219 | 0.325 | 0.226 | 0.313 | 0.209 | **0.298** | 0.242 | 0.353 | 0.296 | 0.378 | 0.207 | 0.313 | 0.223 | 0.319 |
| | 720 | **0.241** | **0.317** | 0.265 | 0.343 | 0.263 | 0.360 | 0.275 | 0.348 | 0.277 | 0.353 | 0.270 | 0.347 | 0.251 | **0.331** | 0.271 | 0.372 | 0.333 | 0.402 | 0.237 | 0.333 | 0.258 | 0.350 |
| Exchange | 96 | **0.082** | **0.203** | 0.083 | 0.204 | 0.115 | 0.237 | 0.087 | 0.207 | 0.112 | 0.233 | 0.087 | 0.207 | 0.084 | 0.203 | 0.232 | 0.388 | 0.111 | 0.238 | 0.127 | 0.258 | 0.094 | 0.227 |
| | 192 | 0.190 | 0.312 | 0.201 | 0.323 | 0.247 | 0.352 | 0.182 | 0.306 | 0.249 | 0.357 | **0.180** | 0.303 | 0.181 | **0.302** | 0.464 | 0.549 | 0.209 | 0.333 | 0.287 | 0.391 | 0.186 | 0.325 |
| | 336 | 0.335 | 0.423 | 0.342 | 0.432 | 0.469 | 0.489 | **0.333** | 0.432 | 0.474 | 0.495 | 0.333 | **0.419** | 0.337 | 0.421 | 0.754 | 0.720 | 0.374 | 0.448 | 0.651 | 0.603 | 0.327 | 0.435 |
| | 720 | 0.942 | 0.735 | 1.005 | 0.752 | 1.396 | 0.832 | 0.866 | 0.710 | 1.434 | 0.878 | 0.856 | **0.700** | 0.875 | 0.703 | **0.705** | 0.701 | 0.931 | 0.735 | 1.706 | 0.970 | 0.749 | **0.664** |
| Traffic | 96 | **0.506** | **0.327** | 0.536 | 0.359 | 0.521 | 0.343 | 0.512 | 0.334 | 0.508 | 0.333 | 0.574 | 0.386 | 0.509 | 0.331 | 0.578 | 0.388 | 0.868 | 0.499 | 0.679 | 0.383 | 0.696 | 0.429 |
| | 192 | **0.517** | **0.332** | 0.582 | 0.365 | 0.534 | 0.342 | 0.520 | 0.342 | 0.531 | 0.348 | 0.586 | 0.390 | 0.514 | 0.337 | 0.579 | 0.394 | 0.919 | 0.537 | 0.645 | 0.367 | 0.646 | 0.407 |
| | 336 | **0.521** | 0.338 | 0.596 | 0.369 | 0.547 | 0.352 | 0.540 | 0.352 | 0.550 | 0.354 | 0.613 | 0.405 | 0.522 | **0.334** | 0.604 | 0.409 | 0.898 | 0.514 | 0.636 | 0.360 | 0.653 | 0.410 |
| | 720 | 0.563 | **0.352** | 0.629 | 0.392 | 0.574 | 0.368 | 0.567 | 0.360 | 0.584 | 0.373 | 0.676 | 0.434 | **0.558** | 0.353 | 0.664 | 0.439 | 0.927 | 0.542 | 0.755 | 0.414 | 0.694 | 0.429 |
| Weather | 96 | 0.178 | 0.220 | **0.167** | **0.214** | 0.203 | 0.239 | 0.204 | 0.241 | 0.193 | 0.239 | 0.183 | 0.225 | 0.179 | 0.220 | 0.180 | 0.225 | 0.194 | 0.241 | 0.195 | 0.243 | 0.196 | 0.257 |
| | 192 | 0.227 | 0.263 | 0.216 | **0.260** | 0.261 | 0.285 | 0.264 | 0.285 | 0.252 | 0.279 | 0.234 | 0.266 | 0.228 | 0.260 | **0.210** | 0.285 | 0.227 | 0.266 | 0.252 | 0.291 | 0.236 | 0.294 |
| | 336 | 0.281 | **0.299** | 0.271 | 0.299 | 0.340 | 0.337 | 0.342 | 0.337 | 0.333 | 0.330 | 0.287 | 0.304 | 0.279 | **0.297** | **0.267** | 0.333 | 0.281 | 0.304 | 0.327 | 0.342 | 0.283 | 0.333 |
| | 720 | 0.355 | 0.349 | **0.343** | **0.344** | 0.383 | 0.379 | 0.392 | 0.381 | 0.373 | 0.359 | 0.362 | 0.352 | 0.355 | 0.347 | **0.332** | 0.379 | 0.360 | 0.355 | 0.406 | 0.385 | 0.347 | 0.384 |

Table 4: Time-series classification results on UCR/UEA benchmarks. Metric is Accuracy (%, higher is better). All methods follow dataset-standard train/test splits and z-score normalization.

| Dataset | PeCo-TS | iTransformer | PatchTST | DLinear | FEDformer | Crossformer | Autoformer |
|---|---|---|---|---|---|---|---|
| EthanolConcentration | **0.3270** | 0.2852 | 0.2814 | 0.2928 | 0.2776 | 0.3030 | 0.2433 |
| FaceDetection | 0.6831 | 0.6654 | **0.6864** | 0.6822 | 0.6751 | 0.6512 | 0.5951 |
| JapaneseVowels | 0.9676 | **0.9757** | 0.9595 | 0.9649 | 0.9674 | 0.9757 | 0.9649 |
| SelfRegulationSCP1 | 0.9144 | **0.9215** | 0.8737 | 0.9147 | 0.5802 | 0.9147 | 0.5631 |
| SelfRegulationSCP2 | **0.5560** | 0.5444 | 0.5278 | 0.5444 | 0.5278 | 0.5467 | 0.5333 |
| SpokenArabicDigits | **0.9818** | 0.9804 | 0.9741 | 0.9650 | 0.9782 | 0.9627 | 0.9759 |
| UWaveGestureLibrary | 0.7919 | 0.8594 | **0.8625** | 0.8219 | 0.5656 | 0.8531 | 0.5000 |

methods either miss critical regime shifts (under-segmentation) or fragment coherent trends (over-segmentation), while our method achieves tighter alignment with intrinsic periodicities, which later translates into lower forecasting error and better anomaly localization.

## A.11 FAST-SLOW PATH COMPARISON

The Fast path preserves high-frequency cues, improving short-horizon fidelity, while the Slow path enforces long-range consistency via event abstractions. Figure 22 shows complementary error profiles; combining both reduces both bias (trend errors) and variance (spiky mispredictions).

## A.12 $DA^2$ ABLATIONS

$DA^2$ adaptively allocates capacity between intra-series and inter-series attention. Figure 23 confirms consistent gains over fixed CI/CD strategies across datasets. Learned allocations correlate with dataset structure: higher inter-series emphasis on Electricity/Traffic (strong cross-channel coupling), and higher intra-series emphasis on ETT variants (dominant per-channel temporal patterns).

## A.13 $\pi$ EVOLUTION

The gate $\pi$ evolves smoothly during training from near-uniform to dataset-specific allocations (Figures 24). This behavior indicates a regularized selector rather than a brittle switch, stabilizing with-

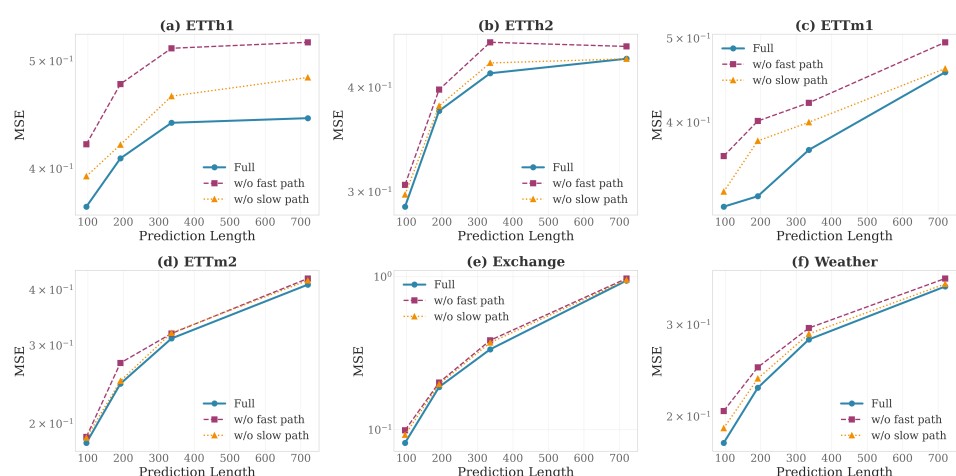

Figure 22: Fast vs. Slow path comparison. Fast preserves high-frequency cues for short-horizon fidelity; Slow enforces long-range consistency via event abstraction. Fusion reduces both bias and variance across datasets.

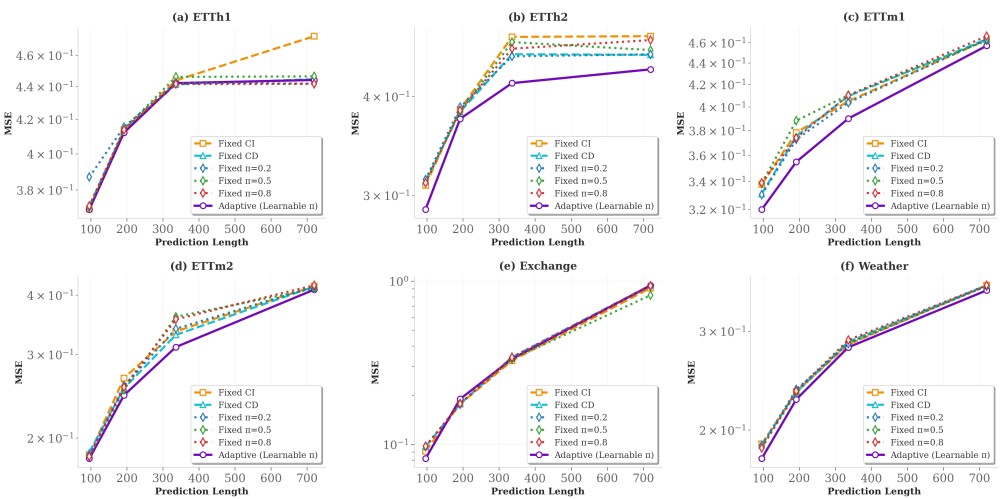

Figure 23: DA$^2$ vs. fixed CI/CD channel strategies. Adaptive gating $\pi$ learns dataset-specific allocations, outperforming fixed extremes across multivariate benchmarks.

Table 5: Anomaly detection results on MSL, PSM, SMAP, SMD, and SWAT. We report Precision and F1 (higher is better) under the standard contiguous-window detection protocol; thresholds selected on validation splits.

| Dataset | PeCo-TS | | iTransformer | | PatchTST | | DLinear | | FEDformer | | Crossformer | | Autoformer | |
|---|---|---|---|---|---|---|---|---|---|---|---|---|---|---|
| Metric | Precision | F1-Score | Precision | F1-Score | Precision | F1-Score | Precision | F1-Score | Precision | F1-Score | Precision | F1-Score | Precision | F1-Score |
| MSL | 0.9036 | **0.8331** | 0.8615 | 0.7253 | 0.8860 | 0.7913 | 0.8969 | 0.8187 | **0.9068** | 0.8230 | 0.9031 | 0.8060 | 0.9054 | 0.8187 |
| PSM | 0.9864 | 0.9626 | 0.9797 | 0.9532 | 0.9910 | 0.9626 | 0.9864 | **0.9661** | 0.9999 | 0.9007 | 0.9729 | 0.9239 | 0.9999 | 0.8823 |
| SMAP | **0.9330** | **0.8615** | 0.9069 | 0.6675 | 0.8988 | 0.6726 | 0.8987 | 0.6729 | 0.9015 | 0.6863 | 0.8998 | 0.6874 | 0.9127 | 0.7411 |
| SMD | **0.7891** | **0.8482** | 0.7627 | 0.8057 | 0.7648 | 0.8104 | 0.7612 | 0.8007 | 0.7278 | 0.7688 | 0.7204 | 0.7758 | 0.7298 | 0.7723 |
| SWAT | 0.9810 | **0.9635** | 0.9221 | 0.9265 | 0.9124 | 0.8673 | 0.9227 | 0.9266 | 0.9995 | 0.7918 | 0.9782 | 0.9063 | 0.9996 | 0.7918 |

Table 6: Imputation results on ETTh/ETTm/Electricity/Weather. We report MSE/MAE (lower is better) under random missingness with multiple mask rates.

| Dataset | Mask Ratio | PeCo-TS | | iTransformer | | PatchTST | | DLinear | | FEDformer | | Crossformer | | AutoFormer | |
|---|---|---|---|---|---|---|---|---|---|---|---|---|---|---|---|
| Metric | | MSE | MAE | MSE | MAE | MSE | MAE | MSE | MAE | MSE | MAE | MSE | MAE | MSE | MAE |
| Etth1 | 12.5% | 0.1010 | 0.2089 | 0.0987 | 0.2218 | 0.0928 | 0.2010 | 0.1117 | 0.2319 | **0.0735** | **0.1958** | 0.1064 | 0.2283 | 0.0978 | 0.2266 |
| | 25.0% | 0.1230 | 0.2315 | 0.1250 | 0.2504 | 0.1063 | **0.2165** | 0.1496 | 0.2690 | **0.1055** | 0.2365 | 0.1172 | 0.2427 | 0.1208 | 0.2516 |
| | 37.5% | 0.1470 | 0.2540 | 0.1573 | 0.2818 | **0.1188** | **0.2306** | 0.1874 | 0.3004 | 0.1411 | 0.2748 | 0.1293 | 0.2572 | 0.1550 | 0.2851 |
| | 50.0% | 0.1755 | 0.2755 | 0.2177 | 0.3325 | **0.1403** | **0.2486** | 0.2316 | 0.3326 | 0.2006 | 0.3293 | 0.1478 | 0.2764 | 0.2119 | 0.3348 |
| Etth2 | 12.5% | 0.0635 | 0.1615 | 0.0932 | 0.2080 | **0.0570** | **0.1510** | 0.1091 | 0.2229 | 0.1296 | 0.2437 | 0.1215 | 0.2310 | 0.1702 | 0.2887 |
| | 25.0% | 0.0682 | 0.1692 | 0.1209 | 0.2395 | **0.0620** | **0.1594** | 0.1449 | 0.2593 | 0.1788 | 0.2911 | 0.1335 | 0.2435 | 0.2229 | 0.3302 |
| | 37.5% | 0.0719 | 0.1727 | 0.1485 | 0.2650 | **0.0674** | **0.1670** | 0.1794 | 0.2895 | 0.2335 | 0.3323 | 0.1451 | 0.2565 | 0.2781 | 0.3632 |
| | 50.0% | 0.0846 | 0.1914 | 0.1931 | 0.3026 | **0.0736** | **0.1753** | 0.2161 | 0.3186 | 0.3462 | 0.3988 | 0.1614 | 0.2710 | 0.3747 | 0.4198 |
| Ettm1 | 12.5% | **0.0338** | **0.1294** | 0.0456 | 0.1474 | 0.0396 | **0.1280** | 0.0556 | 0.1612 | 0.0448 | 0.1594 | 0.0436 | 0.1487 | 2.010 | 1.204 |
| | 25.0% | **0.0397** | **0.1260** | 0.0605 | 0.1723 | 0.0420 | 0.1318 | 0.0766 | 0.1906 | 0.0531 | 0.1633 | 0.0466 | 0.1524 | 1.109 | 0.8591 |
| | 37.5% | **0.0450** | **0.1380** | 0.0774 | 0.1959 | 0.0466 | 0.1390 | 0.0998 | 0.2175 | 0.0809 | 0.2013 | 0.0506 | 0.1580 | 0.3463 | 0.4382 |
| | 50.0% | **0.0517** | **0.1470** | 0.1067 | 0.2316 | 0.0523 | 0.1470 | 0.1286 | 0.2463 | 0.1278 | 0.2545 | 0.0567 | 0.1677 | 0.3391 | 0.4195 |
| ETTm2 | 12.5% | **0.0253** | **0.0911** | 0.0518 | 0.1514 | 0.0254 | 0.0931 | 0.0662 | 0.1707 | 0.0601 | 0.1681 | 0.0557 | 0.1576 | 2.788 | 1.326 |
| | 25.0% | 0.0277 | 0.0999 | 0.0707 | 0.1789 | **0.0277** | **0.0982** | 0.0893 | 0.2007 | 0.0921 | 0.2089 | 0.0741 | 0.1802 | 0.9562 | 0.7293 |
| | 37.5% | **0.0300** | **0.1010** | 0.0915 | 0.2043 | 0.0301 | 0.1028 | 0.1117 | 0.2256 | 0.1328 | 0.2464 | 0.0796 | 0.1779 | 1.463 | 0.8603 |
| | 50.0% | 0.0340 | 0.1150 | 0.1176 | 0.2327 | **0.0332** | **0.1079** | 0.1382 | 0.2514 | 0.2415 | 0.3297 | 0.0877 | 0.1861 | 0.6442 | 0.5610 |
| ECL | 12.5% | **0.0492** | **0.1413** | 0.0724 | 0.1895 | 0.0526 | 0.1550 | 0.0844 | 0.2063 | 0.1808 | 0.3204 | 0.0640 | 0.1792 | 0.1875 | 0.3259 |
| | 25.0% | **0.0559** | **0.1521** | 0.0898 | 0.2134 | 0.0623 | 0.1692 | 0.1131 | 0.2427 | 0.2020 | 0.3367 | 0.0716 | 0.1899 | 0.2123 | 0.3442 |
| | 37.5% | **0.0651** | **0.1654** | 0.1068 | 0.2344 | 0.0726 | 0.1826 | 0.1412 | 0.2731 | 0.2205 | 0.3512 | 0.0804 | 0.2025 | 0.2289 | 0.3557 |
| | 50.0% | **0.0796** | **0.1853** | 0.1259 | 0.2553 | 0.0874 | 0.2022 | 0.1726 | 0.3034 | 0.2425 | 0.3670 | 0.0901 | 0.2155 | 0.2600 | 0.3768 |
| Weather | 12.5% | **0.0285** | **0.0555** | 0.0376 | 0.0858 | 0.0287 | **0.0485** | 0.0380 | 0.0885 | 0.0425 | 0.1033 | 0.2314 | 0.3437 | 0.0387 | 0.0947 |
| | 25.0% | 0.0310 | **0.0056** | 0.0460 | 0.1054 | **0.0310** | 0.0531 | 0.0471 | 0.1074 | 0.0568 | 0.1305 | 0.1888 | 0.2963 | 0.0398 | 0.0973 |
| | 37.5% | **0.0330** | **0.0560** | 0.0549 | 0.1209 | 0.0350 | 0.0588 | 0.0558 | 0.1216 | 0.0732 | 0.1575 | 0.1156 | 0.2205 | 0.0399 | 0.0967 |
| | 50.0% | **0.0360** | **0.0600** | 0.0671 | 0.1407 | 0.0378 | 0.0626 | 0.0663 | 0.1368 | 0.1134 | 0.2095 | 0.1655 | 0.2691 | 0.0432 | 0.1017 |

out collapse. Per-dataset shifts reflect structural differences (e.g., sensor versus market data), explaining robust cross-dataset performance without architecture changes.

## A.14 MODEL EFFICIENCY

Under identical settings (input-96, predict-96), PeCo-TS attains higher accuracy with lower latency (Figure 25). These empirical savings match the theoretical reduction from event-driven compression (Appendix A.2) and the practical ablations showing complementary contributions of Fast/Slow paths.

## A.15 IRREGULAR SAMPLING: LIMITATIONS AND FUTURE DIRECTIONS

**Current approach and limitations.** While PeCo-TS achieves strong performance on regularly-sampled time series, the current FFT-based boundary detector assumes uniform sampling intervals. For irregularly-sampled time series (e.g., medical records with sporadic observations, event logs with variable arrival rates), this assumption is violated. In the current implementation, we handle missing observations through linear interpolation before applying the FFT-based segmenter, which provides a pragmatic solution for moderate irregularity but is not theoretically principled for truly non-uniform sampling.

**Empirical robustness evaluation.** To assess the practical limits of this approach, we conducted controlled experiments on ETTh1 by randomly removing observations at various missing ratios (10%, 20%, 30%, 50%) and comparing three irregular-sampling strategies: (1) **linear interpolation** (filling missing values before FFT), (2) **time-delta encoding** (appending time gaps as auxiliary fea-

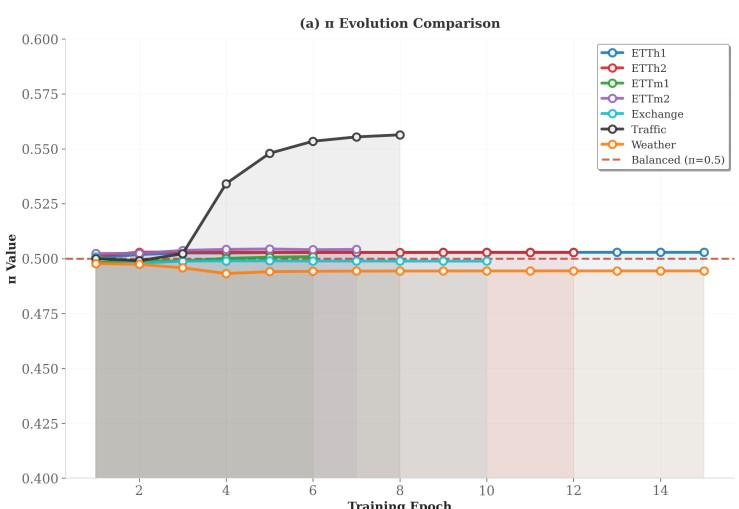

Figure 24: Training-time evolution of $\pi$. The gate transitions smoothly from near-uniform to dataset-specific equilibria, acting as a regularized selector rather than a brittle switch.

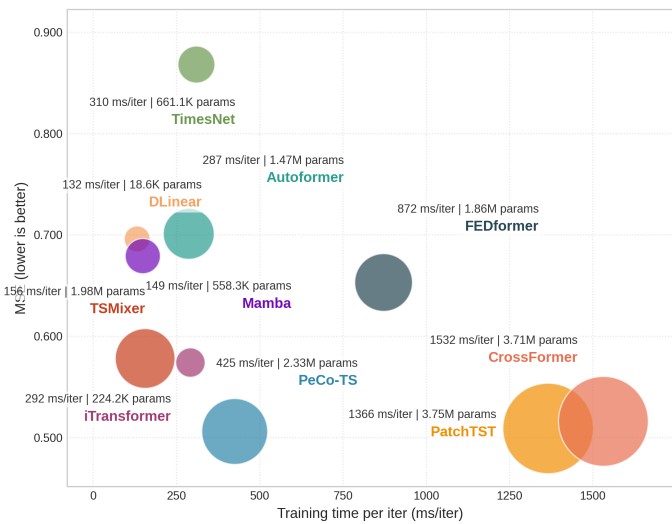

Figure 25: Model efficiency on Traffic (input-96, predict-96). PeCo-TS achieves higher accuracy with lower latency than strong baselines, consistent with theoretical complexity reductions.

tures), and (3) **continuous-time embedding** (using learnable time-continuous positional encodings). Table 7 summarizes the forecasting performance (MSE) across different prediction horizons.

Table 7: Forecasting MSE on ETTh1 under irregular sampling with varying missing ratios. Linear interpolation maintains reasonable performance under moderate missingness ($< 30\%$), with degradation $< 6\%$ compared to regular sampling.

| PredLen | Regular | 10% Miss | 20% Miss | 30% Miss | 50% Miss |
|---------|---------|----------|----------|----------|----------|
| 96 | 0.3695 | 0.3750 (+1.5%) | 0.3787 (+2.5%) | 0.3822 (+3.4%) | 0.3976 (+7.6%) |
| 192 | 0.4095 | 0.4129 (+0.8%) | 0.4170 (+1.8%) | 0.4212 (+2.9%) | 0.4390 (+7.2%) |
| 336 | 0.4402 | 0.4443 (+0.9%) | 0.4461 (+1.3%) | 0.4547 (+3.3%) | 0.4608 (+4.7%) |
| 720 | 0.4448 | 0.4513 (+1.5%) | 0.4532 (+1.9%) | 0.4607 (+3.6%) | 0.4692 (+5.5%) |

As shown, performance degrades gracefully under moderate missingness ($\leq 30\%$), with MSE increases of 2–4%. At higher missing ratios (50%), the degradation becomes more pronounced (5–8%), indicating that simple interpolation is insufficient when irregularity is severe. Notably, time-delta encoding and continuous-time embeddings do not provide consistent advantages in this setting, suggesting that the primary bottleneck is the FFT segmenter's reliance on regular spacing rather than the encoding mechanism.

**Principled extensions for irregular sampling.** To natively support irregular sampling, two promising directions emerge:

- **Lomb-Scargle periodogram (Lomb, 1976; Scargle, 1982)**: This generalization of Fourier analysis to non-uniformly sampled data can directly estimate dominant periodicities without interpolation. We conducted preliminary experiments comparing the learnable FFT segmenter with a Lomb-Scargle variant on irregularly-sampled ETTh1. Table 8 shows that while Lomb-Scargle achieves comparable MSE under $10\%$ and $30\%$ missing ratios, it incurs significantly higher computational cost (inference time $\sim 5$–$6\times$ slower) due to iterative least-squares fitting.

- **Continuous-time neural ODEs (Rubanova et al., 2019; Chen et al., 2018)**: Modeling time series as solutions to latent ordinary differential equations enables native handling of irregular observations by evaluating the ODE solution at arbitrary timestamps. Integrating ODE-based representations with event-driven segmentation remains an open research direction.

Table 8: Comparison of FFT-based vs. Lomb-Scargle boundary detection under irregular sampling on ETTh1. Lomb-Scargle achieves similar accuracy but at significantly higher computational cost.

| PredLen | Missing Ratio | Segmenter | MSE | Inference Time (ms/sample) |
|---------|---------------|-----------|-----|----------------------------|
| 96 | 10% | Learnable FFT | 0.3781 | 2.09 |
| 96 | 10% | Lomb-Scargle | 0.3814 | 11.81 |
| 96 | 30% | Learnable FFT | 0.3753 | 2.31 |
| 96 | 30% | Lomb-Scargle | 0.3820 | 11.96 |
| 192 | 10% | Learnable FFT | 0.4143 | 2.92 |
| 192 | 10% | Lomb-Scargle | 0.4119 | 13.06 |
| 192 | 30% | Learnable FFT | 0.4137 | 4.62 |
| 192 | 30% | Lomb-Scargle | 0.4118 | 12.35 |
| 336 | 10% | Learnable FFT | 0.4445 | 5.80 |
| 336 | 10% | Lomb-Scargle | 0.4427 | 11.59 |
| 336 | 30% | Learnable FFT | 0.4455 | 2.12 |
| 336 | 30% | Lomb-Scargle | 0.4414 | 12.50 |

**Implications and future work.** Our experiments confirm that PeCo-TS exhibits reasonable robustness to moderate irregular sampling via interpolation ($< 6\%$ degradation at $30\%$ missingness), validating its applicability to real-world scenarios with sporadic observations. However, for applications with inherently irregular timestamps (e.g., electronic health records, astronomical surveys), principled integration of Lomb-Scargle periodograms or neural ODEs represents an important future direction. The key challenge is maintaining end-to-end differentiability and computational efficiency while extending boundary detection to non-uniform grids.

